# Spatiotemporal changes in the boreal forest in Siberia over the period 1985–2015 against the background of climate change

Wenxue Fu[1, 2], Lei Tian[1, 2, 3], Yu Tao[4], Mingyang Li[3], Huadong Guo[1, 2]

[1]International Research Center of Big Data for Sustainable Development Goals, Beijing, 100094, China
[2]Key Laboratory of Digital Earth Science, Aerospace Information Research Institute, Chinese Academy of Sciences, Beijing, 100094, China
[3]College of Forestry, Nanjing Forestry University, Nanjing, 210037, China
[4]Anhui Province Key Laboratory of Physical Geographical Environment, Chuzhou University, Chuzhou, 239000, China

*Correspondence to*: Wenxue Fu (fuwx@aircas.ac.cn) and Lei Tian (tianlei@njfu.edu.cn)

**Abstract.** Climate change has been proven to be an indisputable fact and to be occurring at a faster rate (compared to the other regions at the same latitude of the world) in boreal forest areas. Climate change has been observed to have a strong influence on forests; however, until now, the amount of quantitative information on the climate drivers that are producing changes in boreal forest is limited. The objectives of this work were to quantify the spatiotemporal characteristics of boreal forest and forest types and to find the significant climate drivers that are producing changes in boreal forest. The boreal forest in Krasnoyarskiy Kray, Siberia, Russia, which lies within the latitude range 51°N–69°N, was selected as the study area. The distribution of the boreal forest and forest types in the years 1985, 1995, 2005 and 2015 were derived from a series of Landsat data. The spatiotemporal changes in the boreal forest and forest types that occurred over each ten-year period within each 2° latitudinal zone between 51°N and 69°N from 1985 to 2015 were then comprehensively analyzed. The results show that the total area of forest increased over the study period and that the increase was fastest in the high-latitude zone between 63°N and 69°N. The increases in the areas of broad-leaved and coniferous forests were found to have different characteristics. In the medium-latitude zone between 57°N and 63°N in particular, the area of broad-leaved forest grew faster than that of the coniferous forest. Finally, the influence of the climate factors of temperature and precipitation on changes in the forests was analyzed. The results indicate that temperature rather than precipitation is the main climate factor that is driving change.

**Graphical Abstract**

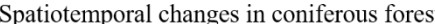

Spatiotemporal changes in coniferous forest

1985-1995   1995-2005   2005-2015   1985-2015

Background of Climate Change

1985   1995   2005   2015

Spatiotemporal changes in broad-leaved forest

1985-1995   1995-2005   2005-2015   1985-2015

Analysis of climatic factors by the PLS-VIP method

## 1 Introduction

Boreal forests occupy between 8% and 11% of the Earth's land surface and store a large fraction of global terrestrial carbon. These forests are found throughout the polar regions of the northern hemisphere (at latitudes of 45°N–70°N) and cover 33% of the total circumpolar region, mainly in the Nordic countries (Finland, Sweden, Norway and Iceland), Russia and North

America (Canada and Alaska) (Allison and Treseder, 2011). The boreal forest biome has one of the largest geographic footprints of any terrestrial biome on the planet (Olson et al. 2001). To date, research into shifts in the range of this biome has predominately focused on the advance of boreal tree species into tundra or alpine habitats (i.e., treeline advance; see Harsch et al. 2009), or on the species-specific responses of temperate tree species (Zhu et al. 2012).

Climate change is expected to lead to changes in temperature and precipitation – factors which strongly influence boreal forests.

In addition, the dynamics of the boreal forest will have a significant impact on global climate–biosphere feedback. Research has indicated that latitudes north of 40° are expected to experience the greatest temperature increases due to climate change (Serreze et al., 2000; Michael et al., 2021). In northern regions, the impacts of climate change are expected to be acute and the

ecological response to climate change will vary spatially (Walther et al., 2002). Changes to biodiversity are one of the expected responses to climate change, for example, some of the most important conifer species in British Columbia are expected to lose

a large portion of their suitable habitat (Hamann and Wang, 2006). These changes will lead to significant spatiotemporal changes in boreal forest. Most importantly, climate change is expected to reduce climatic constraints on plant growth (Nemani et al., 2003): warmer, wetter conditions will result in increased vegetation productivity, which has been demonstrated to be an indirect indicator of biodiversity, correlated with geographic variation in species richness (Coops et al., 2008; Nelson et al., 2014).

There has been much research on the effect of climate change on boreal forest. It has been observed that the growth of boreal forest has been influenced by global warming in the past decade or more. However, there are clear spatiotemporal differences in these effects (Alibakhshi et al., 2020). For example, Hou et al. (2020) found that vegetation phenology indicators in Finland's boreal forests showed spatiotemporal differences in response to climate variables in different months, i.e., vegetation in different regions showed different patterns of response to climate variables. Models and investigations have suggested that

warming will induce northern migration of the treeline and an alteration in the mosaic structure of boreal forests; it has also been shown that, as temperatures increase, white spruce tree growth is declining (Soja et al., 2007). Over the past 30 years, spring and autumn temperatures over northern latitudes have increased by about 1.1 °C and 0.8 °C, respectively (Mitchell and Jones 2005), and the thermal potential growing season has lengthened by about 10.5 days (Barichivich et al., 2013). Several studies indicate that increasing warming may result in accelerating the northward expansion of boreal forests (Veraverbeke et

al., 2017), as well as the observation of a greening trend characterized by a longer growing season and greater photosynthetic activity (Piao et al., 2008). Shuman et al. (2011) showed that climate warming may convert Siberia's deciduous larch (*Larix spp.*) to evergreen conifer forests, and thus decrease regional surface albedo; At the continental scale, when temperature is increased, larch-dominated sites become vulnerable to early replacement by evergreen conifers. Ratcliffe et al. (2017) investigated a forested peatland in western Siberia and showed that climate change has caused the expansion of forested

peatlands and increased tree cover. In addition, it is highly probable that the annual mean temperature in Canada's boreal forest region will increase by at least 2°C by 2050 in this century, which may lead to effects on the ecological functioning of the region's boreal forests, such as triggering a process of forest decline and re-establishment lasting several decades, while also releasing significant quantities of greenhouse gases that will amplify the future global warming trend (Price, et al., 2013). In practice, it is a challenge to quantify the effects of climate change on boreal forest because there are great uncertainties attached

to possible interactions among them, as well as with other land-use pressures (Price et al., 2013). Therefore, the extent of the boreal forest response to climate change is still not fully understood.

A practical method of examining trends in forest cover at large scales is to employ remotely sensed data. Satellite-based monitoring can be implemented consistently across large regions at annual or inter-annual intervals. Time-series of MODIS (Moderate-resolution Imaging Spectroradiometer) satellite data have been commonly used as a data source in many forest

studies. However, the relatively coarse spatial resolution of these data is not sufficient to detect forest-cover changes accurately as it has been shown that a substantial proportion of land-cover changes occur at scales below 250 m (Jia et al., 2014; Heiskanen

et al., 2012). Medium-resolution data, such as that acquired by the Landsat series of sensors including the Thematic Mapper (TM), Enhanced Thematic Mapper Plus (ETM+) and Operational Land Imager (OLI), represent the most widely used multispectral datasets that can be used for monitoring natural and human-induced landscape changes at the scale of tens of meters over periods of years or decades (Matasci et al., 2018; Hadi et al., 2016; Hermosilla et al., 2019). These data have been widely used for forest-cover mapping and change detection because changes in forest cover due to anthropogenic factors usually happen at small scales (Townshend et al., 2012). For example, White, et al. (2017) used the extensive Landsat archive to produce annual, gap-free surface reflectance composites for exploring forest disturbance and recovery characteristics in Canadian boreal forests. Sulla-Menashe, et al. (2018) used normalized difference vegetation index (NDVI) time series from Landsat to explore geographic patterns of greening and browning in Canadian boreal forests, and revealed that continued long-term climate change has the potential to significantly alter the character and function of Canadian boreal forests, with greening observed to be most prevalent in eastern Canada and browning to occur primarily in western Canada.

The objectives of this work were to quantify the spatiotemporal changes occurring in the boreal forests in Siberia and then to find which climate factor was the main driver of these changes. To do this, we tried to answer the following questions. (i) What is the extent of the changes in boreal forest cover and forest types that are occurring? (ii) In which latitude zones are the forest cover and forest types most sensitive to climate change? (iii) Which climate change factor is the main driver influencing change in the boreal forest in Siberia?

In order to answer the questions and meet our objectives, the following work was undertaken.

(1) A typical area of Siberian boreal forest in Krasnoyarskiy Kray, Russia, which extended from the temperate to the frigid zones, was selected as a research area. Forest cover and forest types data for the years 1985, 1995, 2005 and 2015 were retrieved from the Landsat series of imagery.

(2) The characteristics of the spatiotemporal changes in the boreal forest cover and forest types within different latitude zones over the period 1985–2015 were quantified. The results were validated using 987 points sampled from high-resolution Gaofeng-2 satellite images (spatial resolution: 0.81 m) and were found to have an overall accuracy of about 90.37%.

(3) The influences of two climate factors – temperature and precipitation – on changes in boreal forest were analyzed so that the main climate factor driving these changes could be identified.

## 2 Study area

The boreal forest in Krasnoyarskiy Kray in central Russia, which is also located in the middle of Siberia (Figure 1), was selected as the study area. This area extends from approximately 51°N to 69°N and from 84°E to 110°E. The climatic zones found in this area range from temperate in the south to frigid in the north (Brandt 2009), which means that the latitude range was considered sufficiently large for an analysis of the sensitivity of the forest to climate change to be carried out.

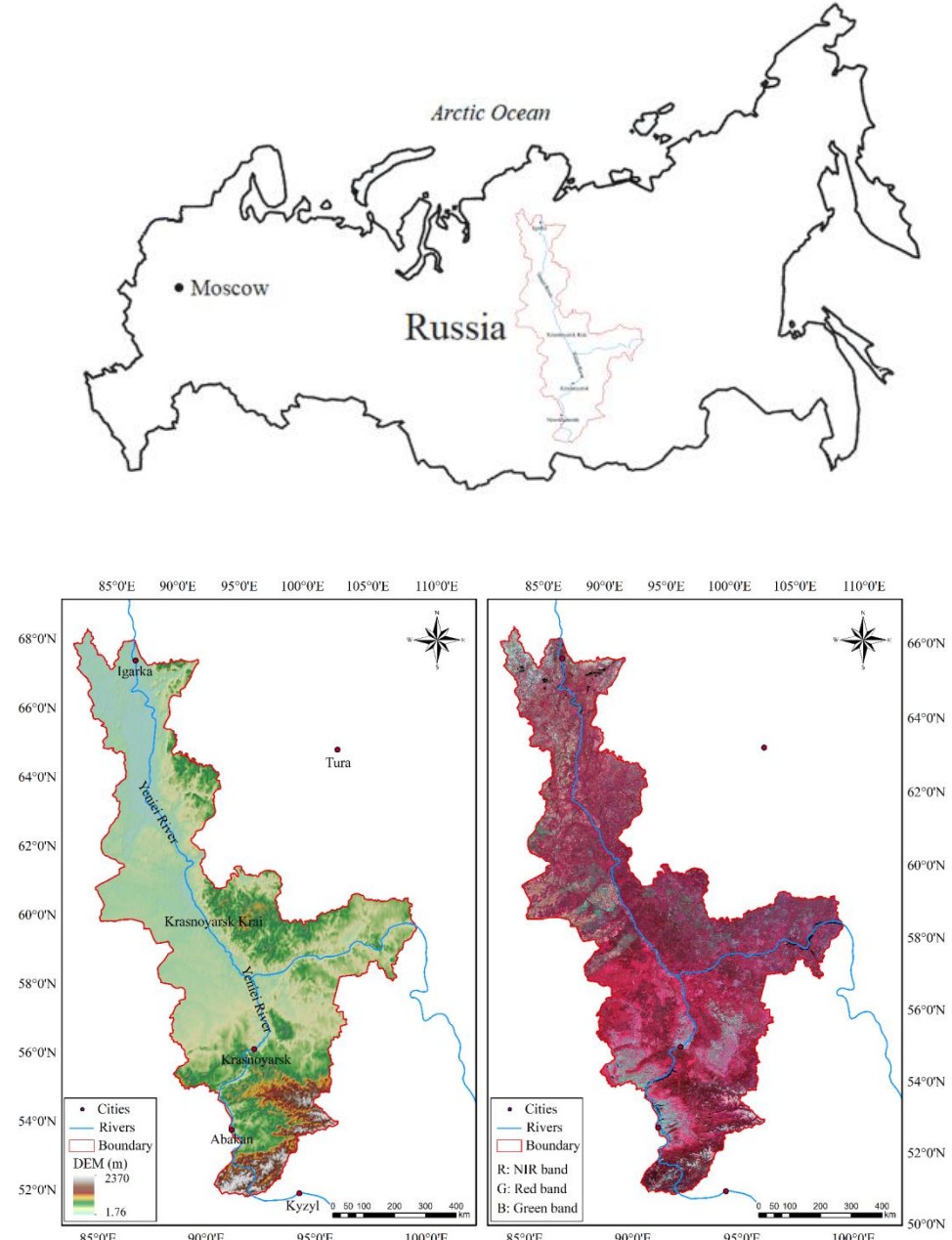

**Figure 1.** Location of the study area together with the DEM and false-color composite of Landsat 8 images.

The climate in the study area is strongly continental with a large temperature gradient from south to north. In the north there are fewer than 40 days each year with temperatures above 10 °C, whereas in the south there are about 110–120 such days. The average temperature in January is −36 °C in the north and −18 °C in the south, and in July 10 °C in the north and 20 °C in the

south. The annual precipitation in the north of the area is 200–300 mm; in the south it is about 1000 mm. The area is sparsely populated; a small number of towns and villages are scattered across the south, surrounded by areas of farmland. We divided the study area into nine latitude zones, each with a width of 2°, from south to north. The area of each of these zones is shown in Table 1.

**Table 1.** List of the latitude zones and their areas.

| Latitude zones (°N) | Area (km$^2$) |
| --- | --- |
| 67−69 | 19596.45 |
| 65−67 | 58813.18 |
| 63−65 | 58364.09 |
| 61−63 | 66364.15 |
| 59−61 | 130507.24 |
| 57−59 | 155232.73 |
| 55−57 | 114456.67 |
| 53−55 | 77579.58 |
| 51−53 | 33564.27 |
| Total | 714478.36 |

## 3 Materials and Methods

### 3.1 Data

More than 300 Landsat Thematic Mapper (TM) and Operational Land Imager (OLI) scenes of the study area containing little to no cloud cover were obtained from the United States Geological Survey (USGS) (http://glovis.usgs.gov/). These images were acquired mainly in the years 1985, 1995, 2005 and 2015. Most of the images were acquired during summer (June to September), three images with no snow over the south area acquired in October and some data from adjacent years were used to make up for the lack of data in the target years. Level 1 Tier 1 data with a spatial resolution of 30 m were used. In addition, four Gaofen-2 (GF-2) satellite panchromatic images acquired in 2015 that had a spatial resolution of 0.81 m were acquired for validation purposes.

The climate datasets ERA5-Land (Hersbach et al., 2020) and ERA-20CM (Hersbach et al, 2015), obtained from the European Centre for Medium Range Weather Forecasts (ECMWF), were used as the source of temperature and precipitation data. The digital elevation model (DEM) data (at 30 m spatial resolution) was obtained from the ASTER GDEM V03 dataset was also used in this study so that the influence of the elevation could be analyzed.

### 3.2 Methods

#### 3.2.1 Landsat data preprocessing

Image preprocessing, including radiometric and atmospheric correction to eliminate radiometric and geometric distortion, was carried out. Following this, a haze optimized transformation (HOT) algorithm was used to identify and remove noise due to thin clouds (Zhang et al., 2002; Li et al., 2019; Liu et al., 2017). Firstly, the clear line was determined according to the high correlation between the blue and red bands in the clear region, and then the HOT value was calculated using the transform equation:

$$HOT = \rho_1 \sin\theta - \rho_3 \cos\theta - |I|\cos\theta \tag{1}$$

where $\rho_1$ and $\rho_3$ are the reflectance of the blue and red bands of TM and OLI images, respectively. $I$ represents the intercept of the clear line, and $\theta$ is the inclination of the clear line.

The cloud images were classified by the HOT value which represented the cloud thickness, and then the Landsat image in cloud region and clear region was automatically classified using just one near-infrared band and two shortwave infrared bands. The image in the cloud region of visible bands was matched to the image in the clear region according to the cloud class and object classification to remove the effect of the cloud (Tian and Fu 2020).

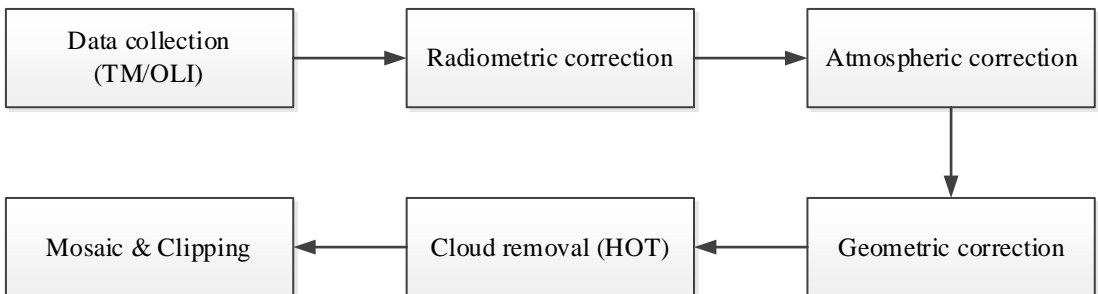

**Figure 2.** Preprocessing of Landsat Thematic Mapper (TM)/Operational Land Imager (OLI) scenes.

#### 3.2.2 Forest cover and types classification

First, a simple decision tree algorithm was used to distinguish vegetation from non-vegetation. In this classification, two vegetation indices, NDVI and ratio vegetation index (RVI), were used for the discrimination between forest and nonforest land. NDVI can effectively weaken the effects of complex terrain in image information extraction, and enhance the distinction between vegetation and other land types, which is helpful for improving the accuracy and credibility of forest information extraction. The RVI can better reflect the difference of vegetation growth and coverage, and is suitable for vegetation monitoring in areas with vigorous vegetation growth and high coverage. The annual maximum NDVI ($NDVI_{max}$) values of built-up areas, barren lands, and sparsely vegetated lands are usually lower than 0.30, whereas forest $NDVI_{max}$ values are

usually higher than 0.50 (Qin et al., 2015). Subsequently, we determined the decision tree classification rules based on sample
training: NDVI values greater than 0.62 and RVI values greater than 6.0 were selected as vegetation land, otherwise land was regarded as non-vegetation land. Next, the areas of vegetation were further classified as being forest or 'other' vegetation. In this work, it was found that different plants have different spectral reflectance peaks in the near-infrared band; this band is highly sensitive to the differences in reflectance that results from different types of leaves having different internal structures and colors (Lewis 2002). Vegetation objects with reflectance values of less than 0.38 in the near-infrared were determined as
being forest land.

Finally, a random forest (RF) algorithm was used to discriminate coniferous and broadleaved forests from areas of vegetation (Breiman 2001; Strobl et al., 2007; Cutler et al., 2008; Svetnik et al., 2003; Rodriguez-Galiano et al., 2013; Assiri 2021; Climent et al., 2019). Representative training samples are one of the most critical components of the RF algorithm. In this study, we selected the sample points used for the classification for different years based on Landsat images refer to GF-2
images and Google Earth images (Gong et al., 2013). Six bands, Landsat TM bands 1–5 and 7, and Landsat OLI bands 2–7, were selected as characteristic spectral variables, and meanwhile NDVI, the normalized difference built-up index (NDBI) (Cha et al., 2003) and the RVI were also selected as index characteristic variables for classification in RF.

### 3.3 Accuracy validation

We selected 987 randomly distributed sampling points from the GF-2 images acquired in 2015 for the accuracy validation.
The overall accuracy was found to be 90.37%, and the F1-scores (Chen et al., 2021; Pontius and Millones, 2011) for the broad-leaved, coniferous forest and non-forest land were 0.85, 0.93 and 0.91, respectively. Considering the consistency of the Landsat series of images, the above validation was still considered to be valid for the earlier years because it is difficult to obtain the measured data or the high-resolution satellite images of the study area for these times.

### 4. Results and discussion

**4.1 Spatiotemporal changes in forest characteristics within the different latitude zones**

First of all, as shown in Figure 3, we analyzed the spatial distribution of the boreal forest within the different latitude zones in 1985, 1995, 2005 and 2015. The overall forest coverage in the study area was high – up to 80.5% – but with significant spatial variations. As the amount of human activity in the study area is limited, these differences can be considered to be caused mainly by natural factors. It was found that the forest coverage within the zones with latitudes in the range 51°N–67°N was
above 60% but that this declined sharply to about 34.4% at 67°N–69°N. The highest rate of forest coverage – about 90% – occurred in the 57°N–59°N zone. In the lower latitude zones in the range 53°N–57°N, the forest coverage was slightly lower as a result of a certain amount of human activity.

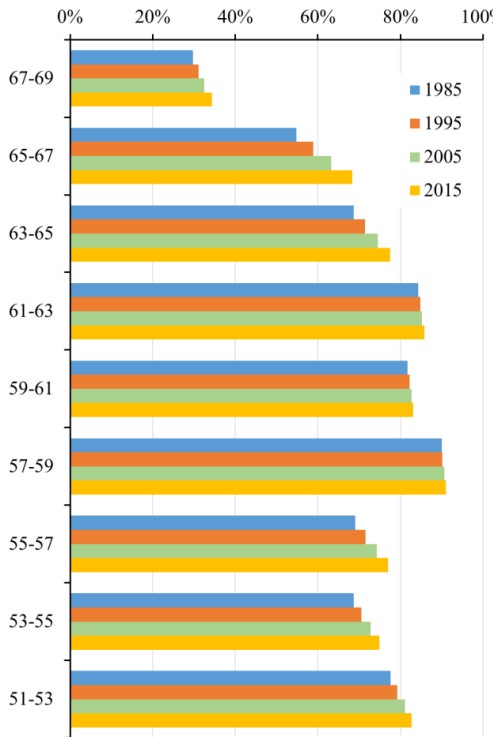

**Figure 3.** Boreal forest land cover within different latitude zones in four times.

An important problem that we focused on was the significant spatiotemporal change in the boreal forest coverage that occurred over the three decades of the study period. We retrieved the changes for the four times intervals of 1985–1995, 1995–2005, 2005–2015 and 1985–2015 – the results are shown in Figure 4. The first three images reflect the changes that occurred over each 10−year interval; the final image shows the overall change over the three decades. Quantitative information about the changes that occurred over the four different time intervals is given in Table 2, and charts detailing the spatiotemporal changes are shown in Figure 5. However, empirical NDVI values for different times and regions are not universal (Ma et al., 2019). Therefore, the results of the threshold-based method (decision tree) used in this study for the classification of forested and non-forested land may have a certain uncertainty. Future research could use the machine learning or deep learning methods to classify forests to improve the accuracy of classification.

Overall, the forest coverage within all of the latitude zones increased continuously over the three decades of the study. We first analyzed the characteristics of the ratio $\Delta R_{fo}$, which is the ratio of the increase in the area of forest to the original area, as well as of $\Delta R_{fl}$, which is the ratio of the increase in the area of forest to the total land area. First, it can be seen that, in the center of the study area from 57°N–63°N, $\Delta R_{fo}$ and $\Delta R_{fl}$ were relatively stable and had average values of less than 2% over the period 1985– 2015. Taking into account the accuracy of the forest cover retrieval, it can be considered that the forest coverage in this zone has not changed over the study period, which means that the cover of boreal forest in this zone has not been significantly

affected by climate change. The rate of forest coverage in this zone was also highest in the study area – more than 80% in some areas – as has been discussed above.

In the other latitude zones, the values of the ratios $\Delta R_{fo}$ and $\Delta R_{fl}$ were significantly higher. The fastest change was observed in the northernmost zone (63°N–69°N) that is also the zone where the climate warming is also projected to be the highest. Over the period 1985–2015, the average value of $\Delta R_{fo}$ was about 17%; for $\Delta R_{fl}$, it was about 9%. The highest rates of increase in forest cover occurred in the zone 65°N–67°N, where $\Delta R_{fo}$ and $\Delta R_{fl}$ were about 24.61% and 13.50%, respectively, which is equivalent to average annual values of about 0.76% and 0.44%. Finally, in the southern zone from 51°N–57°N, the average value of $\Delta R_{fo}$ was about 9% and of $\Delta R_{fl}$ about 6%.

There were also temporal variations in the changes in forest coverage, especially in the high-latitude zones. For example, between 65°N and 69°N, the amount of forest coverage showed an accelerating trend over the study period, with average values of $\Delta R_{fo}$ and $\Delta R_{fl}$ of 6.00% and 2.70%, respectively, for the time interval 1985–1995 and 7.00% and 3.50%, respectively, for 2005–2015. However, the rates of increase in forest coverage in the other latitude zones were relatively stable over the study period.

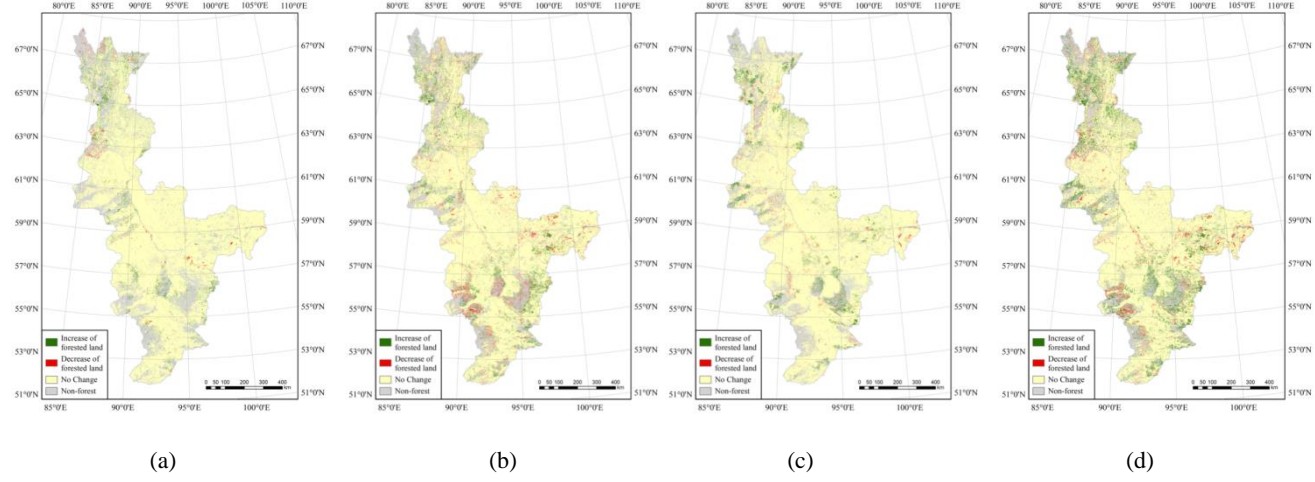

(a)  (b)  (c)  (d)

**Figure 4.** Changes in boreal forest coverage in the time intervals (a) 1985–1995, (b) 1995–2005, (c) 2005–2015 and (d) 1985–2015.

**Table 2.** Quantitative details of the changes in boreal forest coverage in each latitude zone.

| Latitude (°N) | 1985–1995 | | 1995–2005 | | 2005–2015 | | 1985–2015 | |
|---|---|---|---|---|---|---|---|---|
| | $\Delta R_{fo}$ | $\Delta R_{fl}$ | $\Delta R_{fo}$ | $\Delta R_{fl}$ | $\Delta R_{fo}$ | $\Delta R_{fl}$ | $\Delta R_{fo}$ | $\Delta R_{fl}$ |
| 51–53 | 2.06% | 1.60% | 2.35% | 1.86% | 1.93% | 1.56% | 6.47% | 5.02% |
| 53–55 | 2.82% | 1.93% | 3.12% | 2.20% | 2.94% | 2.14% | 9.14% | 6.28% |
| 55–57 | 3.50% | 2.42% | 3.84% | 2.75% | 3.76% | 2.79% | 11.52% | 7.96% |

| 57–59 | 0.21% | 0.19% | 0.45% | 0.40% | 0.46% | 0.41% | 1.11% | 1.00% |
|-------|-------|-------|-------|-------|-------|-------|-------|-------|
| 59–61 | 0.57% | 0.47% | 0.62% | 0.51% | 0.47% | 0.39% | 1.67% | 1.37% |
| 61–63 | 0.63% | 0.53% | 0.54% | 0.46% | 0.68% | 0.58% | 1.86% | 1.57% |
| 63–65 | 4.03% | 2.77% | 4.30% | 3.07% | 4.03% | 3.00% | 12.87% | 8.84% |
| 65–67 | 7.47% | 4.10% | 7.18% | 4.23% | 8.17% | 5.16% | 24.61% | 13.50% |
| 67–69 | 4.45% | 1.32% | 4.43% | 1.38% | 5.70% | 1.85% | 15.28% | 4.55% |

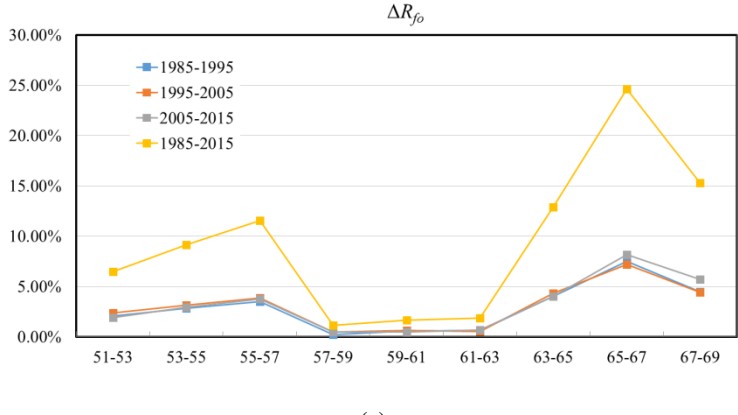

(a)

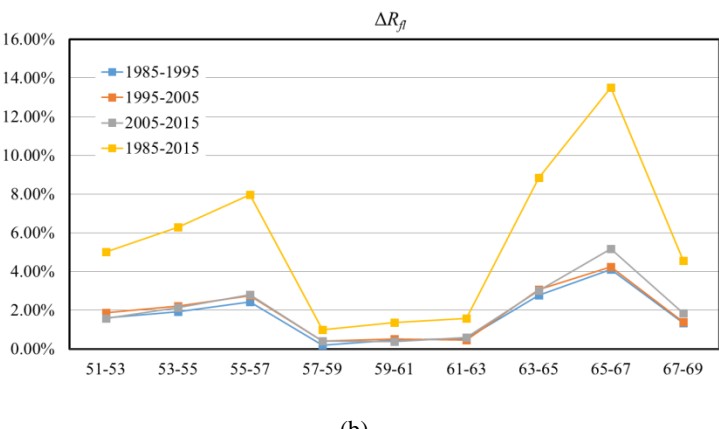

(b)

**Figure 5.** Forest coverage changes in different latitude zones during different time intervals as measured by (a) $\Delta R_{fo}$ and (b) $\Delta R_{fl}$.

### 4.2. Spatiotemporal characteristics of changes in forest types

**4.2.1 Spatiotemporal differences in the ranges of forest types**

We also analyzed the spatiotemporal changes in the broad-leaved and coniferous forest coverage. $R_{bf}$ was defined as the ratio of the broad-leaved forest area to the total forest area; the corresponding ratio for coniferous forest, $R_{cf}$, was then given by $1 - R_{bf}$. Similarly, $R_{bl}$ was used to represent the ratio of the broad-leaved forest area to the total land area; the corresponding

measure for coniferous forest was denoted as $R_{cl}$. Quantitative information about the forest types coverage can be found in
Table 3, and the differences between these ratios are shown in Figure 6.

It can be seen from Figure 6(a) that the region most suited to coniferous forest within the study area is the medium latitude
zone 57°N–63°N, which has a value of $R_{cl}$ of about 70%. In the northernmost zone (67°N–69°N), $R_{cl}$ is still above 25% whereas
$R_{bl}$ is only about 5%, which indicates that coniferous forest is more resistant to cold and that broad-leaved forest is essentially
not found north of latitude 67°N in the studied region.

Broad-leaved forest cover is low in the north and high in the south; the highest $R_{bl}$ value – over 40% –occurs in the 53°N–
57°N zone; however, this falls sharply to less than 20% at latitudes above 57°N. In the 51°N–57°N zone, there may be some
human activity that affects the forest coverage, which leads to the slightly lower values of $R_{bl}$ and $R_{cl}$.

Broad-leaved forest co-exists with coniferous forest across the whole study area, and the proportions of the two types of forest
are similar in the 53°N–57°N zone (Figure 6(c)). However, in the other zones, coniferous forest is the dominant species and
235 $R_{cf}$ is above 70%.

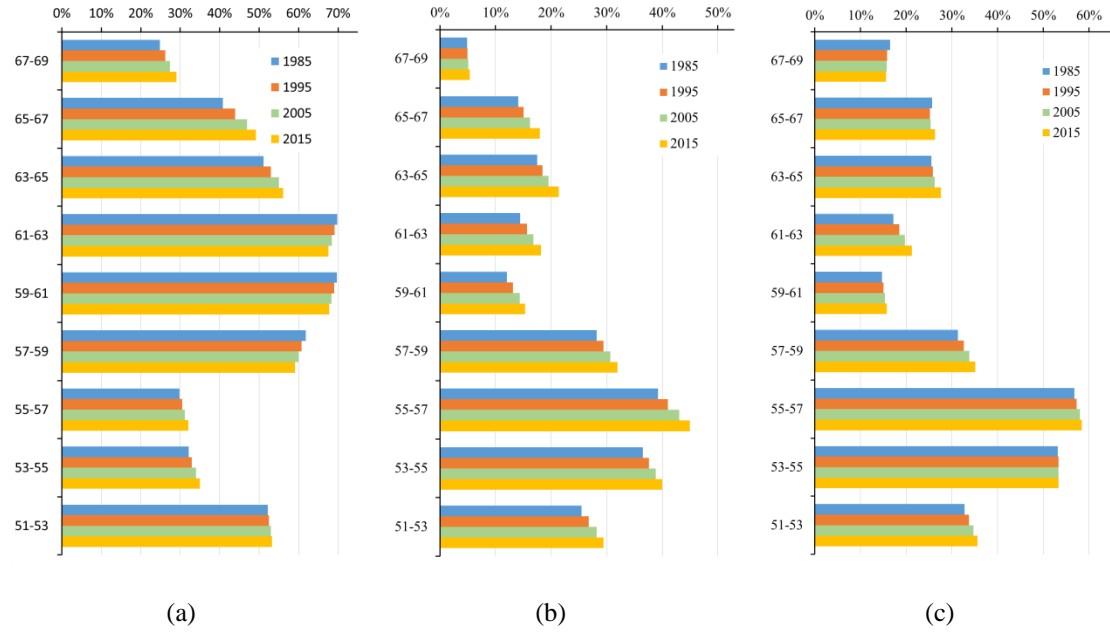

(a)                          (b)                          (c)

**Figure 6.** Differences between the rates of broad-leaved and coniferous forest coverage as measured by (a) $R_{cl}$, (b) $R_{bl}$ and (c) $R_{bf}$.

**Table 3.** Quantitative details of the coverage of different forest types.

| Latitude (°N) | 1985 | | | 1995 | | | 2005 | | | 2015 | | |
|---|---|---|---|---|---|---|---|---|---|---|---|---|
| | $R_{bf}$ | $R_{bl}$ | $R_{cl}$ | $R_{bf}$ | $R_{bl}$ | $R_{cl}$ | $R_{bf}$ | $R_{bl}$ | $R_{cl}$ | $R_{bf}$ | $R_{bl}$ | $R_{cl}$ |
| 51–53 | 32.79 % | 25.46 % | 52.18 % | 33.76 % | 26.75 % | 52.49 % | 34.75 % | 28.18 % | 52.92 % | 35.59 % | 29.42 % | 53.25 % |

| | | | | | | | | | | | | |
|---|---|---|---|---|---|---|---|---|---|---|---|---|
| 53–55 | 53.17 % | 36.50 % | 32.15 % | 53.31 % | 37.63 % | 32.96 % | 53.36 % | 38.84 % | 33.95 % | 53.35 % | 39.98 % | 34.96 % |
| 55–57 | 56.78 % | 39.22 % | 29.85 % | 57.32 % | 40.98 % | 30.51 % | 58.02 % | 43.07 % | 31.17 % | 58.41 % | 44.99 % | 32.04 % |
| 57–59 | 31.34 % | 28.20 % | 61.79 % | 32.61 % | 29.41 % | 60.77 % | 33.84 % | 30.65 % | 59.93 % | 35.08 % | 31.92 % | 59.07 % |
| 59–61 | 14.71 % | 12.03 % | 69.71 % | 16.02 % | 13.17 % | 69.03 % | 17.34 % | 14.34 % | 68.37 % | 18.48 % | 15.36 % | 67.74 % |
| 61–63 | 17.18 % | 14.47 % | 69.77 % | 18.47 % | 15.66 % | 69.12 % | 19.74 % | 16.83 % | 68.41 % | 21.23 % | 18.22 % | 67.60 % |
| 63–65 | 25.50 % | 17.51 % | 51.17 % | 25.85 % | 18.47 % | 52.97 % | 26.21 % | 19.53 % | 54.99 % | 27.61 % | 21.41 % | 56.11 % |
| 65–67 | 25.68 % | 14.08 % | 40.76 % | 25.50 % | 15.03 % | 43.91 % | 25.64 % | 16.20 % | 46.98 % | 26.29 % | 17.97 % | 50.37 % |
| 67–69 | 16.47 % | 4.91% | 24.89 % | 15.87 % | 4.94% | 26.18 % | 15.72 % | 5.11% | 27.39 % | 15.59 % | 5.35% | 29.00 % |

## 4.2.2 Spatiotemporal characteristics of changes in forest types

The characteristics of the changes in forest types were also analyzed. The spatiotemporal characteristics of these changes for the time intervals 1985–1995, 1995–2005, 2005–2015 and 1985–2015 are shown in Figure 7 and Figure 8, and related quantitative information is shown in Table 4. We denoted $\Delta R_{bl}$ and $\Delta R_{cl}$ as representing the ratios of the change in broad-leaved and coniferous forest coverage to the total land area, respectively, meaning that these are measures of the absolute increase in these forest types. $\Delta R_{bf}$ and $\Delta R_{cf}$ denote the increase in the ratio of the area of broad-leaved forest and coniferous forest to the total area of forest, respectively, meaning that these are measures of the relative increase in the area of forest; $\Delta R_{cf} = -\Delta R_{bf}$.

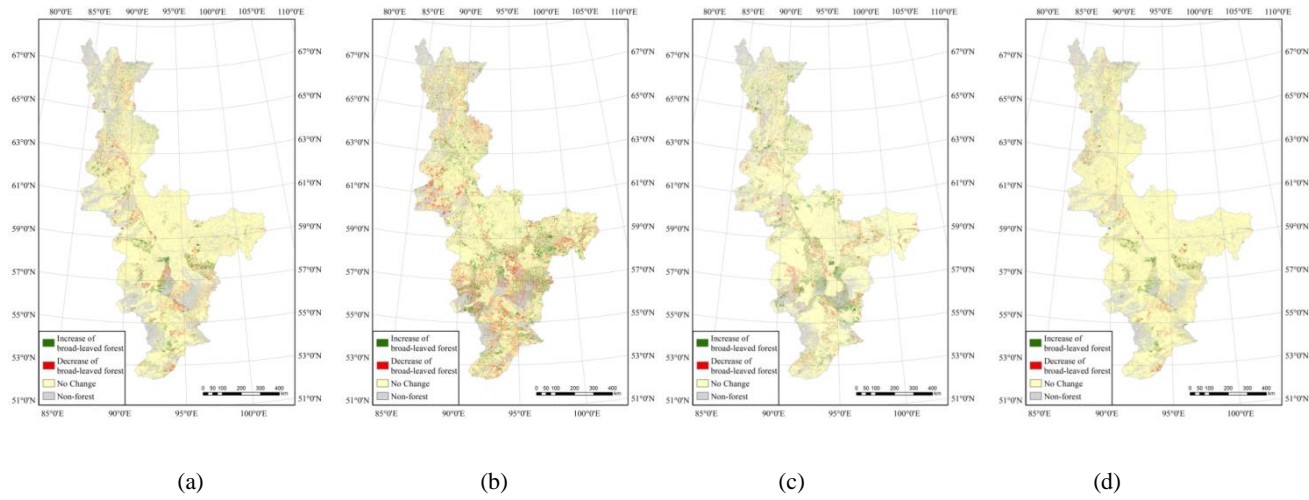

(a)  (b)  (c)  (d)

**Figure 7.** Changes in broad-leaved forest area for (a) 1985–1995, (b) 1995–2005, (c) 2005–2015 and (d) 1985–2015.

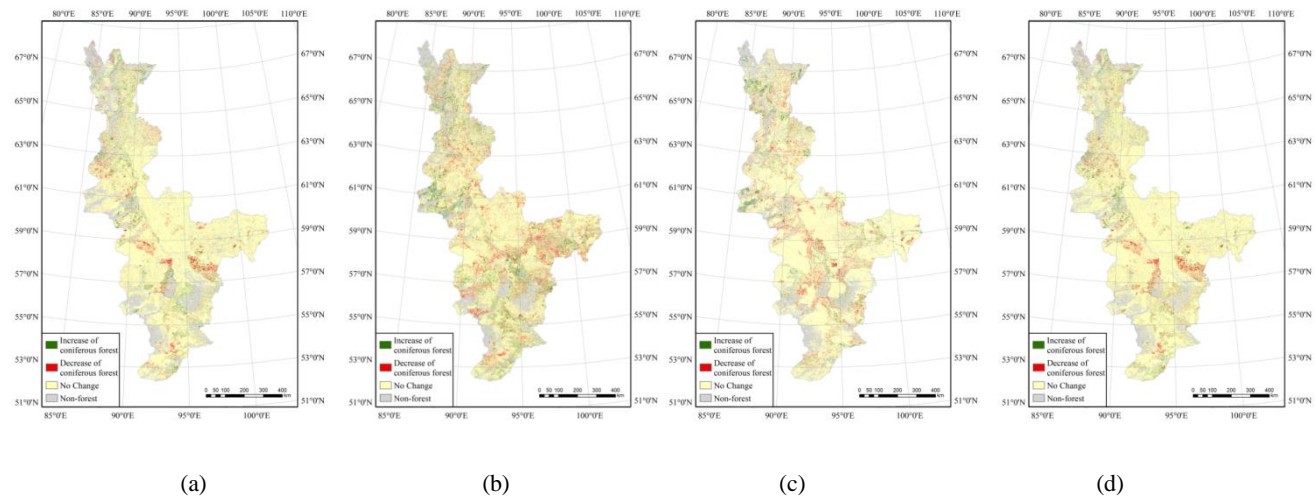

(a)  (b)  (c)  (d)

 **Figure 8.** Changes in coniferous forest area for (a) 1985–1995, (b) 1995–2005, (c) 2005–2015 and (d) 1985–2015.

**Table 4.** Quantitative details of the changes in forest types.

| Latitude | 1985–1995 | | | 1995–2005 | | | 2005–2015 | | | 1985–2015 | | |
|---|---|---|---|---|---|---|---|---|---|---|---|---|
| (°N) | $\Delta R_{bl}$ | $\Delta R_{cl}$ | $\Delta R_{bf}$ | $\Delta R_{bl}$ | $\Delta R_{cl}$ | $\Delta R_{bf}$ | $\Delta R_{bl}$ | $\Delta R_{cl}$ | $\Delta R_{bf}$ | $\Delta R_{bl}$ | $\Delta R_{cl}$ | $\Delta R_{bf}$ |
| 51–53 | 1.29% | 0.31% | 0.97% | 1.43% | 0.43% | 0.99% | 1.24% | 0.33% | 0.84% | 3.96% | 1.06% | 2.79% |
| 53–55 | 1.13% | 0.81% | 0.14% | 1.21% | 0.99% | 0.05% | 1.14% | 1.00% | −0.01% | 3.47% | 2.80% | 0.18% |
| 55–57 | 1.76% | 0.66% | 0.54% | 2.09% | 0.66% | 0.69% | 1.92% | 0.87% | 0.39% | 5.77% | 2.19% | 1.63% |
| 57–59 | 1.21% | −1.02% | 1.28% | 1.24% | −0.84% | 1.22% | 1.27% | −0.86% | 1.24% | 3.72% | −2.72% | 3.74% |
| 59–61 | 1.14% | −0.68% | 1.31% | 1.17% | −0.66% | 1.32% | 1.02% | −0.63% | 1.15% | 3.33% | −1.97% | 3.77% |

| 61–63 | 1.19% | −0.65% | 1.29% | 1.17% | −0.71% | 1.27% | 1.39% | −0.81% | 1.48% | 3.74% | −2.18% | 4.05% |
|---|---|---|---|---|---|---|---|---|---|---|---|---|
| 63–65 | 0.96% | 1.81% | 0.35% | 1.06% | 2.01% | 0.36% | 1.88% | 1.13% | 1.40% | 3.89% | 4.95% | 2.11% |
| 65–67 | 0.95% | 3.15% | −0.18% | 1.17% | 3.06% | 0.14% | 1.77% | 3.39% | 0.65% | 3.89% | 9.61% | 0.62% |
| 67–69 | 0.03% | 1.29% | −0.60% | 0.17% | 1.21% | −0.15% | 0.24% | 1.61% | −0.14% | 0.45% | 4.11% | −0.89% |

We analyzed the spatial characteristics of the changes in forest types, and details of the spatiotemporal changes in these types are shown in Figure 9. Overall, it can be seen that the broad-leaved forest coverage increased in every latitude zone, which means that the climate change that has been occurring may have promoted the growth of broad-leaved species across the study area during the three decades of the study. In the 51°N–67°N zone, the overall value of $\Delta R_{bl}$ is above 3.3%; however, the value of this measure declines rapidly to about 0.45% north of 67°N, which can be considered as being equivalent to there being no change in $R_{bl}$ in this area. The largest value of $\Delta R_{bl}$ over the three decades of the study was 5.77%, which occurred in the latitude zone 55°N–57°N; this is equivalent to an average annual increase of 0.19% and indicates that the broad-leaved forest in this zone was the most sensitive to climate change and its area increased the fastest.

However, coniferous forest showed different change characteristics from those of broad-leaved forest. The latitude zones in the study area can clearly be divided into three parts according to the characteristics of the changes in the coniferous forest area. First, in the zone 51°N–57°N, the average value of $\Delta R_{cl}$ is about 2.0%. However, the area of coniferous forest in the medium latitude zone 57°N–63°N has declined slightly over the three decades of the study with a value of $\Delta R_{cl}$ of about −2.3%; in comparison $\Delta R_{bl}$ is about 3.9%, which means that climate change may have had a negative impact on coniferous forest growth in this zone. The area of coniferous forest increased relatively rapidly in the northern zone between 63°N and 69°N with an average $\Delta R_{cl}$ greater than 4.0%. The largest value of $\Delta R_{cl}$ – 9.61% – occurs in the latitude zone 65°N–67°N, which is equivalent to an average annual increase of 0.32%; for comparison, this zone has a value of $\Delta R_{bl}$ of 3.89%.

It should be noted that $\Delta R_{bf}$ is an important parameter for evaluating the relative proportions of the two-forest types and can reflect the different characteristics of their responses to climate change. A positive $\Delta R_{bf}$ indicates that the proportion of broad-leaved forest relative to the total forest area is increasing faster than that of coniferous forest and that broad-leaved forest is tending to become the dominant tree species. There are a variety of evidence points to complex connections (and changes) in the relationship between disturbance regimes and climate change in boreal forest (Kasischke and Turetsky 2006; Balshi et al., 2009; de Groot et al., 2013). In particular, studies have found that warming and drying trends in Canada's boreal regions favor higher frequency of both fire and insect disturbance (Sulla-Menashe, et al., 2018). While in Siberia, Warming has led to an increase in the frequency and area of wildfires that have reached the Arctic Ocean shore, which is the most important factor in taiga dynamics; furthermore, larch and Scots pine have evolved under conditions of periodic forest fires, thereby gaining a competitive advantage over non-fire adapted species (Kharuk et al., 2021), which may affect forest cover and forest types change in the region. It can be seen from Figure 9(c) that, in the zone 57°N–63°N, $\Delta R_{bf}$ is above 3.8%. Meanwhile, as discussed above, the absolute increase given by $\Delta R_{cl}$ is negative whereas $\Delta R_{bl}$ is positive in this zone. Also, as $R_{bf}$ had a value of 35.08% in 2015 in the zone 57°N–59°N, at the current rate of change, broad-leaved forest will replace coniferous forest as the dominant

tree species in this zone in about 120 years. In general, species will be more resilient at the centers of their present-day distributions, while changes in succession and species composition will be most rapid at the boundaries. Based on current knowledge, the boreal climate zones are expected to shift 5–10 times faster than the speed of natural range expansion achievable by most tree species (McLachlan et al., 2005; McKenney et al., 2007; Aitken et al., 2008; Loarie et al., 2009).

The changes in the area of broad-leaved forest exhibit considerable variations with time, especially in the zone from 61°N–67°N, where the increase in the area of broad-leaved forest accelerated over the three decades of the study. It can be seen from Figure 9(a) that, in the 65°N–67°N zone, $\Delta R_{bl}$ reached 1.93% in the period 2005–2015, whereas it was 1.17% in 1995–2005 and 0.79% in 1985–1995. The results for $\Delta R_{bf}$ in this zone show similar trends, indicating that broad-leaved forest is highly sensitive to climate change and that the increase in its area has been accelerating. Previous studies have shown that early northward colonization of tundra ecozones may be dominated by black and white spruces, which are often already established at the treeline. Where soil conditions permit (or where they are improving as a result of warming and drying), air-borne seeds from birch and aspen are likely to arrive and germinate success fully, leading gradually to a forest with significantly greater deciduous content (Price et al., 2013). However, in the 51°N–61°N and 67°N–69°N zones, the values of $\Delta R_{bl}$, $\Delta R_{cl}$ and $\Delta R_{bf}$ are relatively stable, which shows that the rate of increase in these forest types did not change much over the period studied. Therefore, the key to the validity of the response of boreal forests to climate change is to determine whether climate warming is driving significant expansion beyond the present-day forest extent, or faster stand growth and replacement (Zhu et al., 2013).

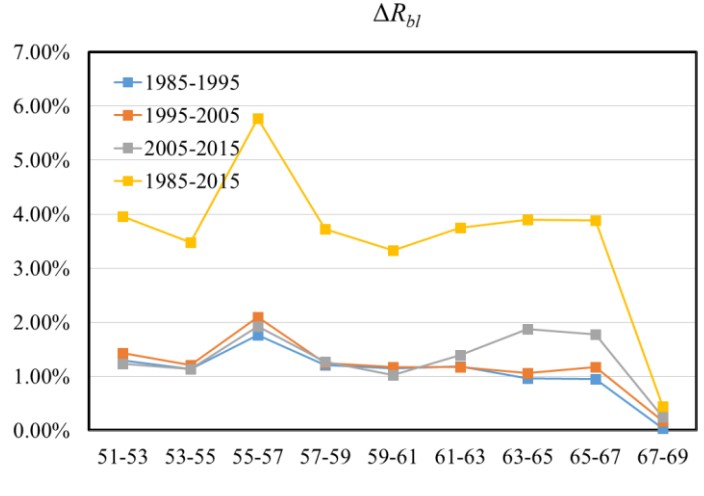

(a)

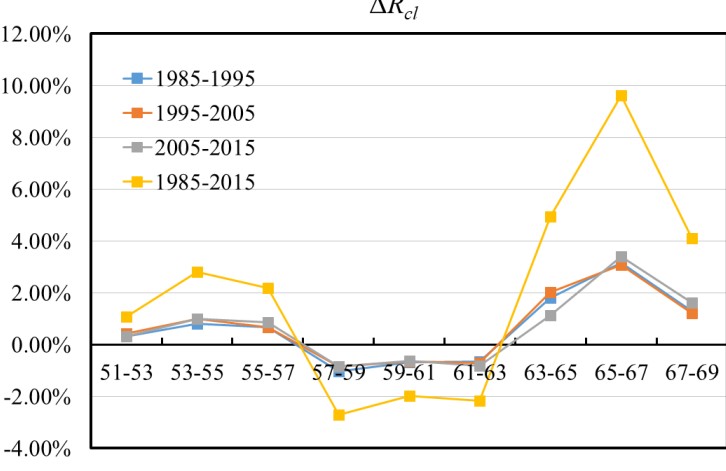

(b)

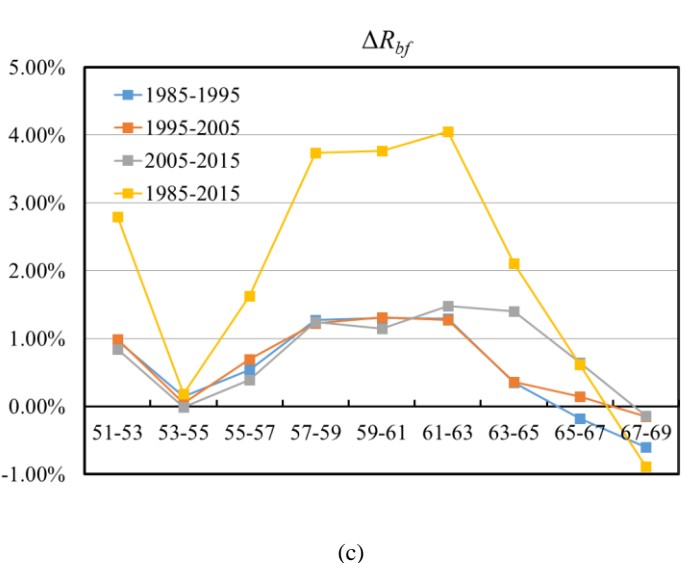

(c)

**Figure 9.** Changes in different forest types in with different latitude zones: (a) $\Delta R_{bl}$, (b) $\Delta R_{cl}$ and (c) $\Delta R_{bf}$.

### 4.3 Analysis of the influences of climate factors on changes in the boreal forest

Given that the amount of human activity in the study area is limited, it is reasonable to assume that the changes in the forest may be driven mainly by climate variables. Two climate factors, temperature and precipitation, were analyzed to find which was the main driving factor behind these changes. The raster climate data were clipped and resampled. The average temperature and total precipitation for the year 2000 are shown in Figure 10. It can be seen that, on the whole, both the temperature and precipitation were lower in the high-latitude zone than in the low-latitude zone.

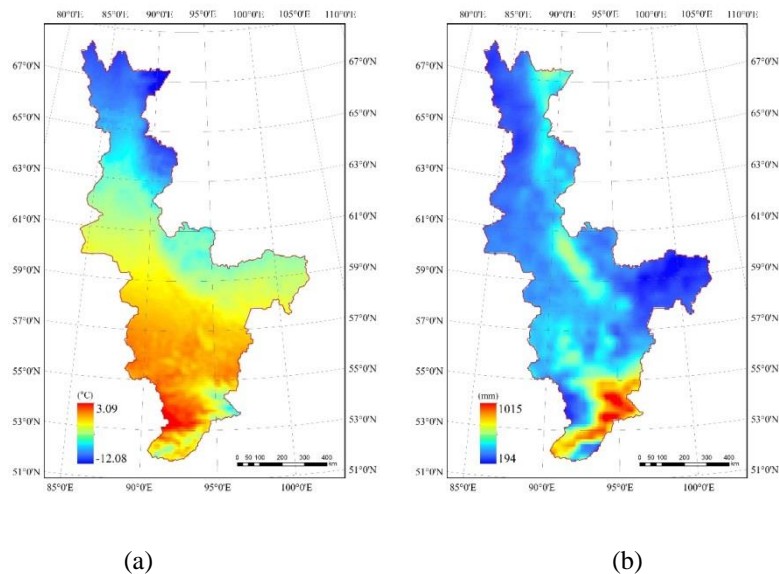

(a)                                                      (b)

**Figure 10.** (a) Average temperature and (b) total precipitation in the study area for the year 2000.

Considering the delay response of the forest to the climate change, the climate data of the time intervals 5–year, 10–year and
15–year before the forest change were chosen for the influence analysis respectively, to determine which the driving interval
was. The climate variables used in this study included the average annual temperature (TEM_Year), the growing season
temperature (TEM_Grow: the average temperature during the growing season (June to September)), the average maximum
temperature during the growing season (TEM_Growmax), the average minimum temperature during the growing season
(TEM_Growmin), the total annual precipitation (PRE_Year) and the total precipitation during the growing season
(PRE_Grow), which have also been used in some previous researches (Hou, et al.,2020, He, et al., 2020). The statistics of
climate variables in the study area from 1971 to 2015 are shown in Figure 11.

The partial least squares (PLS) regression method was adopted for analyzing the effect of the climate factors on forest cover
and changes in forest types. The PLS method is a robust multivariate technique that combines features of principal component
analysis and multiple regression (Abdi 2010); this makes it more parsimonious and statistically robust than principal
component regression (Smoliak et al., 2015). Moreover, since the selected climate variables in our study were to some extent
interrelated, this could have led to multicollinearity, which occurs whenever an in-dependent variable is highly linearly
correlated with one or more other independent variables. PLS regression can effectively deal with the problem of overfitting
that results from multicollinearity (Hou et al., 2020). Thus, PLS regression was particularly suitable for application in our case.
In addition, the variable importance in projection (VIP) score was used to estimate the importance of each independent variable
in the PLS regression – the VIP score represents the statistical contribution of each independent variable to the overall fitted
PLS regression model across all latent vectors (Matthes et al., 2015). A higher VIP score for an independent variable indicates
that the variable is more important in explaining the volatility of the dependent variable(s), and independent variables with a

VIP score greater than 1 are considered significant (Chong et al., 2005). The cross-validated $R^2$ value, which is the square of the correlation between the actual and predicted values, is often called $Q^2$ in PLS regression analysis. A PLS component can be kept and is considered statistically significant in the regression model if its $Q^2$ value is greater than or equal to 0.0975 (Abdi 2010).

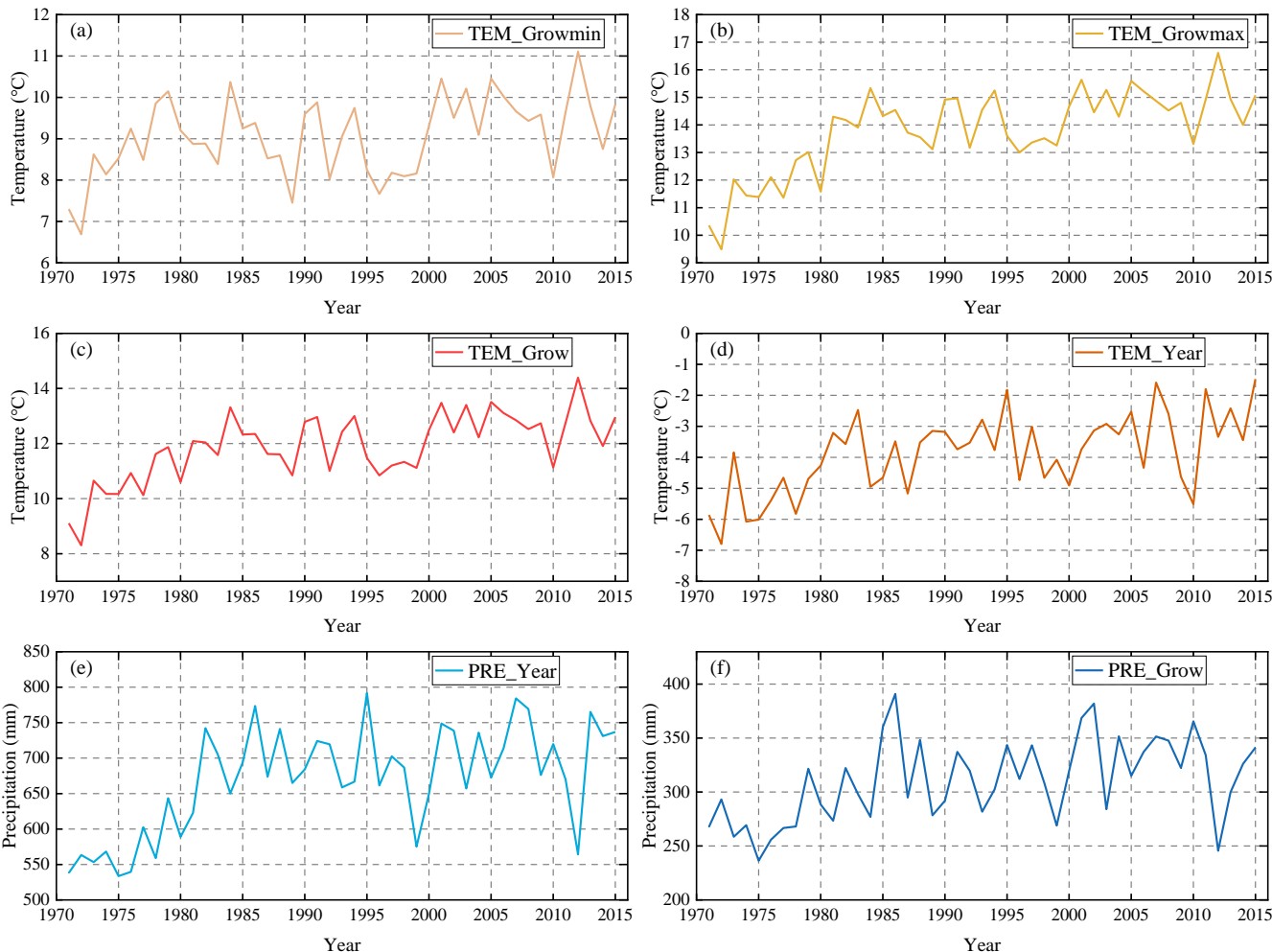

**Figure 11.** Statistics of temperature and precipitation in study area from 1971 to 2015: (a) average minimum temperature during the growing season, (b) average maximum temperature during the growing season, (c) average temperature during the growing season, (d) average annual temperature, (e) total annual precipitation and (f) total precipitation during the growing season.

Using the PLS–VIP method, VIP scores were calculated for use in the interpretation of the contribution of each climate variable to changes in the forest cover and forest types in the study area. The VIP scores for the six climate variables were ranked in descending order. The independent variable with the highest VIP score was considered to be the most important variable (Hou et al., 2020). To clearly show the effect of each climate variable on the forest cover and forest types for each time interval (5 years, 10 years or 15 years), we also calculated the standardized correlation coefficients (Table 5) and $Q^2$ value for each PLS

regression model. The $Q^2$ values for the effect of the climate variables on the forest cover and forest types changes in the PLS regression models for the 5–year, 10–year, and 15–year time intervals were 0.22, 0.21, and 0.20, respectively. This indicates that the climate variables for the 5–year intervals were slightly better correlated with the changes in forest cover and forest types and that they explain the changes in the boreal forest the best. Therefore, a 5–year time interval was selected for subsequent analysis of the response of the forest to climate change.

**Table 5.** The VIP scores of climate variables for different time intervals and their corresponding standardized regression coefficients.

| Climate Variables | 5–year time interval | | | | 10–year time interval | | | | 15–year time interval | | | |
|---|---|---|---|---|---|---|---|---|---|---|---|---|
| | VIP score | Coef_$\Delta$ $R_{fo}$ | Coef_$\Delta$ $R_{fl}$ | Coef_$\Delta$ $R_{bl}$ | VIP score | Coef_$\Delta$ $R_{fo}$ | Coef_$\Delta$ $R_{fl}$ | Coef_$\Delta$ $R_{bl}$ | VIP score | Coef_$\Delta$ $R_{fo}$ | Coef_$\Delta$ $R_{fl}$ | Coef_$\Delta$ $R_{bl}$ |
| TEM_Year | 1.32 | −0.19 | −0.11 | 0.16 | 1.29 | −0.19 | −0.11 | 0.15 | 1.29 | −0.17 | −0.10 | 0.15 |
| TEM_Grow | 1.29 | −0.18 | −0.10 | 0.15 | 1.29 | −0.19 | −0.11 | 0.15 | 1.25 | −0.17 | −0.09 | 0.14 |
| TEM_Growmax | 1.06 | −0.15 | −0.08 | 0.13 | 1.12 | −0.16 | −0.09 | 0.13 | 1.20 | −0.16 | −0.09 | 0.14 |
| TEM_Growmin | 1.16 | −0.16 | −0.09 | 0.14 | 1.14 | −0.16 | −0.10 | 0.13 | 1.08 | −0.15 | −0.08 | 0.12 |
| PRE_Year | 0.02 | 0.00 | 0.00 | -0.00 | 0.03 | −0.00 | −0.00 | 0.00 | 0.05 | −0.01 | −0.00 | 0.01 |
| PRE_Grow | 0.32 | −0.05 | −0.03 | 0.04 | 0.39 | −0.06 | −0.03 | 0.04 | 0.38 | −0.05 | −0.03 | 0.04 |

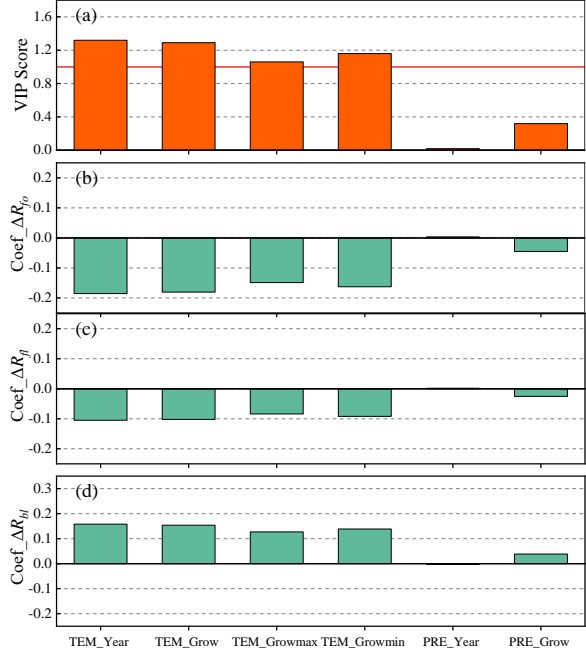

**Figure 12.** Relationship between the forest cover and forest types changes and climate variables for 5–year intervals obtained using PLS regression models: (a) VIP scores; (b), (c), (d) show the corresponding standardized regression coefficients.

Based on the PLS correlation results, we tried to identify the dominant climate variables causing changes in the boreal forest for 5–year intervals. As illustrated in Figure 12(a), TEM_Year is the most significant variable, followed by TEM_Grow,

TEM_Growmax and TEM_Growmin. This indicates that temperature is the main climate factor influencing the boreal forest cover. This result is consistent with previous studies which have demonstrated that temperature is the key factor controlling vegetation growth in most areas of the northern hemisphere (Chen et al.,2018; Menzel et al., 2006; Wang et al., 2011). According to the regression coefficients, $\Delta R_{fo}$ and $\Delta R_{fl}$ have negative responses to all the temperature variables although the correlation coefficients vary (Figure 12(b) and (c)). In other words, an increase in temperature least to a decrease in the total forest coverage. Unexpectedly, precipitation was not found to be a significant variable in terms of boreal forest change (Figure 12(a)). This may be because increasing temperatures affect the response of vegetation to other climate variables (e.g., precipitation). For example, since the mid-1990s, the absence of summer precipitation associated with rising temperatures has had a negative impact on the greenness of boreal forests in western central Eurasia (Buermann et al., 2014). It may also be the case that forest growth is not affected by the drought stress caused by insufficient precipitation – however, this depends on the ability of the forest to access soil water and local microsite conditions (Nicolai-Shaw et al. 2017).

In addition, it can be seen from Figure 12(d) that $\Delta R_{bl}$ responds positively to an increase in temperature, which indicates that broad-leaved forest is sensitive to warming and benefits from higher temperatures. Higher temperature-induced permafrost degradation has severely altered vegetation dynamics in boreal forest areas, notably across the vast areas of northern peatlands and taiga (Price, et al., 2013). A warming climate provides more favorable conditions for the growth and regeneration of plants that previously experienced harsh conditions, thus allowing broad-leaved forest species to survive in the cold north of the study area where they were previously unable to grow (Hoonyoung Park et al, 2015; Høgda et al., 2013; Salminen and Jalkanen, 2015). This pattern has also been observed in Finland (Kauppi et al., 2014). Moreover, it has been found that the distribution of many forest types is beginning to expand towards the poles, with temperate forests gradually shifting into areas previously covered by boreal forests; the southern parts of these temperate forests will be replaced by subtropical or tropical forests. The northern boundary of temperate forests is also shifting towards the poles (Hirota et al., 2010; Parmesan and Yohe, 2003). Similarly, the effect of precipitation on $\Delta R_{bl}$ was found to be insignificant; however, the precipitation variables are not negligible in terms of the response of boreal forests to climate change as this response is a result of the effects of different interrelated climate variables (Hou et al., 2020). Furthermore, warming has a positive effect on $\Delta R_{bl}$, meaning that the observed increase in $\Delta R_{bl}$ was may be caused by a rise in temperatures. The increased area of broad-leaved forest may be due to broad-leaved forest growing in areas that were previously coniferous or not-forest land – further quantitative investigation into the details of these changes is still needed (Hermosilla et al., 2019). Finally, it can be seen from Figure 12(c) and (d) that $\Delta R_{fl}$ and $\Delta R_{bl}$ have opposing responses to the climate factors that were investigated, which means that, where the expansion in the total forest cover is limited, the increase in boreal forest may be mainly due to changes in tree species within the forest (see Figure 7(d) and Figure 8(d)). Several recent studies in Canada have also indicated that the response of forest covers to climate warming varies by tree species (McManus et al., 2012; Weijers et al., 2018). Additionally, forest fires and climate are interrelated, and increasing temperatures and potential decreases in precipitation possibly increase the frequency of wildfires in Siberia, which will inevitably result in changes in forest cover and forest types dynamics (kuaruk et al., 2021). Most importantly, given the projected rate of climate change in the Siberian boreal forest, continued research is necessary to more fully understand how

future changes in temperature and precipitation regimes in the boreal region will affect coupled patterns of forest cover and forest types change in this vulnerable, geographically extensive biome.

**5. Conclusion**

In this study, changes in the area of the Siberian boreal forest and the forest types in Krasnoyarskiy Kray, Russia, were quantified using remote sensing data covering the period 1985 to 2015. The results show that there are differences in the changes that were observed across the study area. Overall, the total forest area increased continuously over the three decades of the study, particularly in the high-latitude part of the study area, which may indicate that the boreal forest in this region is the most sensitive to climate change. It was also found that the changes in broad-leaved and coniferous forest differ according

to latitude. At the medium-latitude zone between 57°N and 63°N, the rate of increase in the area of broad-leaved forest was faster than that of coniferous forest, which means that there is a trend towards broad-leaved forest replacing coniferous forest as the dominant tree species in the future.

Overall, it was found that an increase in temperature tends to inhibit the expansion of the forest; however, an increase can promote the growth of broad-leaved forest. The effect of precipitation on the total forest area and types of forest species present

was found to be negligible. In addition, the influence of anthropogenic factors may affect the response of the forest to climate change and, to some degree, reduce the effects of climate change. High spatial resolution data (e.g., WorldView, GF-2 data) have the potential to provide accurate information about vegetation at a fine scale; however, the use of such data may be limited by the low temporal resolution and the extent of cloud cover over Siberia.

**Financial support.** This work was supported by the Innovative Research Program of the International Research Center of Big Data for Sustainable Development Goals (No. CBAS2022IRP03); the Strategic Priority Research Program of the Chinese Academy of Sciences (XDA19070102); the National Natural Science Foundation of China (No. 61971417) and the Innovation Drive Development Special Project of Guangxi (GuikeAA20302022).

**Competing Interests.** The authors declare that they have no known competing financial interests or personal relationships that could have appeared to influence the work reported in this paper.

**Data Availability.** All Landsat Thematic Mapper (TM) and Operational Land Imager (OLI) data of the study area openly available from the United States Geological Survey (USGS) (http://glovis.usgs.gov/). The climate datasets ERA5-Land are

included in Hersbach et al. (2020), https://doi.org/10.1002/qj.3803. The climate datasets ERA-20CM are included in Hersbach et al. (2015), https://doi.org/10.1002/qj.2528.

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
