# Peer review of "Spatiotemporal changes in the boreal forest in Siberia over the period 1985–2015 against the background of climate change"

_Earth System Dynamics, 2022_

## Referee Comment (RC2)

Peer-review of manuscript ESD-2022-18 submitted to Earth System Dynamics

The manuscript by Fu et al., entitled "Spatiotemporal changes in the boreal forest in Siberia over the period 1985–2015 against the background of climate change" presents a change analysis in forest cover over a broad area in Central Siberia. The changes in forest are then examined in the context of climate change. The analysis is based on time series of Landsat TM images that are validated with high resolution images from a satellite instrument named Gaofen-2. The specific research questions ask 1) what the extent of change in forest cover and proportions of tree species are, 2) at what latitude forest cover and tree species are most sensitive to climate change, and 3) which climate change factor is the main driver influencing the observed changes. The main findings state that the total forest cover increased over the study period, with coniferous and broadleaved forests showing different patterns of change at different latitudes. The authors identified the forest changes are driven mostly by temperature instead of precipitation.

As the authors state, global change is especially affecting and driving changes in the high latitude boreal forests. These changes are less often studied in boreal forests of northern Eurasia compared to, e.g., North America. Hence, the premise of the study is interesting and the topic merits examination. The study covers a broad area, and the quality of the technical work is very high. However, I do not think that all the conclusions of the study are credible. This concern is especially related to research question three that examines the connection between the climatic variables and forest changes. In my opinion, the changes observed in forest cover and the proportion of deciduous and coniferous forest are too easily attributed to climate change. Furthermore, the role of natural forest dynamics and disturbances is completely disregarded in the manuscript although they could have a strong explanatory power on the observed changes. I think the results need to be interpreted also in this context! I also think that the quality of the study could be increased with a more thorough examination of the used remote sensing data. For example, what is the proportion of non-forest land turning into forest dominated by broadleaved forest during the study period, what influence do major disturbances such as fire have on the results? I think the material used in this study (i.e. Landsat images) would also enable considering the role of forest disturbances. As a second major concern, I think that the text needs a thorough revision to enhance its readability. These and my other concerns are described in more detail below.

Major comments:

1. I am not convinced that the observed increase in the proportion of broadleaved forest is solely driven by changes in temperature – correlation does not necessarily mean causation. The natural succession of Siberian boreal forests typically follows a pathway where broadleaved trees or *Larix spp.* dominate the early successional stage. In later successional stages the broadleaved trees with short longevity are often replaced by conifers such as *Pinus spp.* and *Picea spp*. if stand-replacing disturbances remain absent. The disturbance regime of the area examined in this study includes both ground and crown fires, the latter occurring especially in young coniferous forests that also have woody debris that acts as a fuel load. According to Kharuk et al. 2021 (https://doi.org/10.1007/s13280-020-01490-x) and fire maps complied by Global Forest Watch, large fires have occurred in the studied area, even during the study period. I would be interested to know what is the role of these disturbances and forest dynamics in, e.g., explaining the observed increases in the proportion of broadleaved trees? Similarly, cessation of anthropogenic land use, such as abandonment of agricultural land and clearcutting without reforesting may start a natural succession and increase the proportion of broadleaved trees (as hinted by the authors at L. 325). I think the role of these processes should be examined prior stating that the observed changes are driven by climate change. Maybe you could use the Landsat images to quantify the rate at which non-forest land converted to broadleaved or conifer forest during the study period, and check whether disturbances could explain some of this conversion (see https://doi.org/10.1016/j.rse.2019.111403)? Forest fire and climate are also interlinked as rising temperatures and potentially declining precipitation may increase the fire prevalence in Siberia. This could also be considered in the paper.

2. How were coniferous and broadleaved trees separated in the study? According to the paragraph title this should be explained in the paragraph starting at L. 110, but I do not find the information from therein. According to Hovi et al. 2017 (https://doi.org/10.14214/sf.7753), *Larix spp.* and certain deciduous trees have similar spectral characteristics. Is it possible that certain *Larix spp.* stands were falsely classified as broadleaved trees, influencing the obtained results?

3. I would be interested to know if the observed changes in forest cover are due to forests with open canopy structure becoming denser or conversion of previously non-forested land into forests? These two processes have very different significance for forest dynamics. Including results of this examination in the study would increase its information content and general interest.

Minor comments:

L. 1 (title): In the manuscript text the authors state that the study area ranges from temperate to boreal forests. If that is the case, I suggest rephrasing the title of the study.

L. 10: At a faster rate compared to where?

L. 11 – 12: I would argue that there is quite a lot of evidence on how the climate change is changing boreal forests. This view is shared by the authors as the introduction at L. 45 states that "There has been much research on the effect of climate change on boreal forest". I think that in this context there has been limited focus on Central Russian boreal forests. Please rephrase the introduction.

L. 15: At first, I did not know what was meant by the term "forest species", but after reading on I understood that they refer to broadleaved and coniferous forests. I would not call these "forest species" but forest types. I would change "forest species" to "forest type" and explicitly say that by forest type you mean coniferous and broadleaved forests.

L. 31 – 32: What is "geographical footprint"? Largest area?

L. 32: Remove "and encircles the globe at northern latitudes" as circumpolar distribution is already mentioned at L. 30.

L. 32 – 35: I find the ending of this sentence paradoxical. It reads that research of range shifts in boreal forests has focused on species-specific responses on temperate tree species (i.e. different biome). Please rephrase what is meant by this.

L. 40 – 41: Could you be a bit more specific – what kind of changes in biodiversity are expected due to climate change?

L. 42 – 44: The message of this sentence is very difficult to understand. Please clarify the point that the sentence tries to make.

L. 46 – 47: Could you give an example of the spatiotemporal differences in tree growth in the boreal biome that are caused by climate change?

L. 49: White spruce (*Picea glauca*) is a species that is native to boreal North America. Changes in growth of white spruce are then hardly relevant in the context of Eurasian boreal forest, right? Could this be replaced with an example from the same study region?

L. 59 – 61: This statement is not true. See, e.g., https://doi.org/10.1007/s10980-020-00979-w on the use of multispectral aerial photographs for this purpose.

L. 70 – 71: What is meant by "from the temperature to the frigid zones"?

L. 83 (Fig. 1): An inset map would be helpful in locating the study region.

L. 111 – 118: Why is discriminating forested/vegetated areas from non-vegetated areas presented twice in these paragraphs?

L. 126: Did you visually classify the sampling points based on GF-2 images?

L. 153 – 154: That the forest cover has not changed much over the study period does not mean that the forests are not significantly affected by climate change but that the influence does not manifest as changes in forest cover. Please rephrase.

L. 157 – 158: Similar to the previous comment, from the fact that the strongest change in forest cover was observed in the northernmost zone it does not follow that the forests in these areas were mostly affected by climate change (see also my comment on forest dynamics). Rephrase "The fastest change was observed in the northern zone from 63°N–69°N, which means that this is the area where the forest has been most affected by climate" for example to "The fastest change was observed in the northernmost zone (63°N–69°N) that is also the zone where the climate warming is also projected to be the highest".

L. 187: Change to "not found north of the latitude 67°N in the studied region.

L. 202: Change "increase" to "change" as the forest cover may also decrease, right?

L. 217 – 219: As I have stated in my previous comments, the fact that the coverage of broadleaved forests has increased in the study region is not necessarily only due to climate change. Please revise.

L. 228 – 229: Also other drivers than climate change may explain the decrease in the cover of coniferous forest. Please revise.

L. 254 – 255: See the previous two comments.

L. 318 – 320: For me it is uncertain if the author suggest that the results indicate a northwards shift in the range of temperate biome. Such range shifts occur during longer time scales than those considered in this study. Please clarify what is meant by this sentence.

L. 319: Replace "temperature" with "temperate".

L. 320: Are you suggesting that the southernmost forests of the study region are transforming into subtropical or tropical forests? Please rephrase.

L. 324: Also other drivers than climate change may explain the identified patterns.

L. 333 – 334: The term "significant difference" refers to differences that are statistically significant. However, the significance of the changes in forest cover was not statistically tested in the study. Please rephrase.

L. 335 – 336: That the forest cover changed increased the most in the northernmost latitudes does not automatically mean that this region is the most sensitive to climate change. Instead, these forest might be – for example – recovering from a major disturbance. Please revise.

---

## Author Comment (AC1)

**Response to Anonymous Referee #1 Comments**

Dear Editors and Referee:

Thanks for your kind comments concerning our manuscript entitled "Spatiotemporal changes in the boreal forest in Siberia over the period 1985–2015 against the background of climate change". Those comments are all valuable and very helpful for improving our paper, and we have made correction which we hope meet with approval. Revised portion are marked in the manuscript. The main corrections in the paper and the responds to the reviewer's comments are as flowing:

Anonymous Referee #1:

The study is comprehensive with relevant details regarding research focus, methods, results, and concluding remarks presented. However, the background of the research focus could be improved to highlight the novelty of the present study. Some parts of the manuscript need improvements.

Introduction

Line 50: "Therefore, the extent of the boreal forest response to climate change is still not fully understood". The author should point out why? What current research has been done, what are the important methods, and what main conclusions are related to this aspect?

**Reply:** Thanks for your suggestion. Based on your suggestion, we now added the current status of research on the response of boreal forests to climate change and also pointed out the existing challenges and main conclusions. (Line 49-60)

Over the past 30 years, spring and autumn temperatures over northern latitudes have increased by about 1.1 °C and 0.8 °C, respectively (Mitchel and Jones 2005), and the thermal potential growing season has lengthened by about 10.5 days (Barichivich et al., 2013). Several studies indicate that increasing warming may result in accelerating the northward expansion of boreal forests (Veraverbeke et al., 2017), as well as the observation of a greening trend characterized by a longer growing season and greater photosynthetic activity (Piao et al., 2008). In addition, it is highly probable that the annual mean temperature in Canada's boreal forest region will increase by at least 2°C by 2050 in this century, which may lead to effects on the ecological functioning of the region's boreal forests, such as triggering a process of forest decline and re-establishment lasting several decades, while also releasing significant quantities of greenhouse gases that will amplify the future global warming trend (Price, et al., 2013). In practice, it is a challenge to quantify the effects of climate change on boreal forest because there are great uncertainties attached to possible interactions among them, as well as with other land-use pressures (Price et al., 2013). Therefore, the extent of the boreal forest response to climate change is still not fully understood.

Line 60: I guess there are studies that have used Landsat to quantify the spatiotemporal changes occurring in the boreal forests in Siberia or other places. It would be nice to see more backgrounds related to this here.

In general, the introduction needs to be improved and the contextual description is weak.

**Reply:** We have added background to the introduction on the use of Landsat data to quantify spatial and temporal variability in boreal forests and made it more readable. (Line 71-76)

For example, White, et al. (2017) used the extensive Landsat archive to produce annual, gap-free surface reflectance composites for exploring forest disturbance and recovery characteristics in Canadian boreal forests. Sulla-Menashe, et al. (2018) used normalized difference vegetation index (NDVI) time series from Landsat to explore geographic patterns of greening and browning in Canadian boreal forests, and revealed that continued long-term climate change has the potential to significantly alter the character and function of Canadian boreal forests, with greening observed to be most prevalent in eastern Canada and browning to occur primarily in western Canada.

Study area

Adding a layer of DEM and Landsat RGB images for the study area is necessary.

**Reply:** Thanks. According to your suggestion, we have added DEM and Landsat RGB images layers for the study area. (Line 96-97)

[Figure]

**Figure 1.** Location of the study area together with the DEM and false-color composite of Landsat 8 images.

Input data

How many tiles of Landsat images could cover the study area?

**Reply:** About 40 Landsat images are required to cover the study area.

"Mainly in the years 1985, 1995, 2005, and 2015", Why did you use this ten years interval? Why did you choose level1 data? USGS also provides level-2 datasets that have been atmosphere corrected, which could significantly reduce the workload in data processing. Why are most of the images acquired from June to September, and what about the others? Please specify them!!

**Reply:** Thanks for this comment. On the one hand, changes in forests, especially in forest types, may require 10 years or even decades to be observed, and on the other hand, limited by Landsat

data availability, thus we chose ten years interval data for this study.

Level 1 Tier 1 data with a spatial resolution of 30 m were used in this work. Landsat scenes with the highest available data quality are placed into Tier 1 and are considered suitable for time-series analysis. Tier 1 includes Level-1 Precision and Terrain (L1TP) corrected data that have well-characterized radiometry and are inter-calibrated across the different Landsat instruments. On this basis, we performed a series of pre-processing of the Level 1 Tier 1 data, including radiation correction, atmospheric correction and HOT algorithm to remove cloud noise (Figure 2), to ensure that the processed data could well support the subsequent forest cover and species classification. Therefore, this dataset could well support the monitoring of dynamic changes of boreal forest cover and species in this study. Certainly, we support your point of view that the atmosphere corrected level 2 dataset can significantly reduce the workload in data processing. We would consider using this dataset for the future research.

[Figure]

**Figure 2.** Preprocessing of Landsat Thematic Mapper (TM)/Operational Land Imager (OLI) scenes.

In this work, most of the images were acquired in summer seasons, mainly between June and September. Due to cooler temperatures in the other months of the study area, there may be snowfall or yellowing and shedding of broadleaf forest leaves, etc., which may lead to accuracy of forest cover and species classification. Therefore, the image time was chosen in summer. Several data acquired in October are used to make up the images due to data deficiency in the current years. In fact, there were only 3 scenes in October, and over relatively warm areas. We also checked these images with no snow. The information has been added in this manuscript. (Line 110-111)

Line 102 What is the spatial resolution of ERA5-land?
**Reply:** The ERA5-land data with a resolution of 0.1° × 0.1°, i.e., native resolution is 9 km.

Data processing
The description of the data processing is very shallow. Given that spatial and temporal change is the key part of this paper, how to proceed with cloud, anomalies, and alignment correction of images is very critical.
**Reply:** According to your suggestion, we have added a detailed introduction to the pre-processing of Landsat data, including radiometric correction, atmospheric correction, image cloud removal, etc. The information has been added in this manuscript. (Line 123-133)

Forest cover and species classification
Why not use random forest classification directly, but use rule-based classification first, and then random forest? I am very skeptical about the final classification result. Due to the large latitudinal differences in the study area, using NDVI threshold classification may result in some forests being removed incorrectly. In addition, there will be subtle differences in NDVI values over the years.

Anyway, the threshold classification method should be used with caution in applying large spatial scales and change detection. How many training samples were used for the classifications? Did you use the same training samples for different years?

**Reply:** Thanks for your careful suggestion. We used a hierarchical classification method for forest cover and forest species, i.e., we first used decision trees to distinguish between forested and non-forested land, and then used a random forest approach to classify forest species. This can effectively avoid the influence on the final classification results due to same object with different spectrums and same spectrum of different objects (e.g., grassland and cultivated land with very similar spectral profiles to forest).

We agree with you that the NDVI values have subtly varied over the years. We have checked the effect of NDVI thresholds on the classification of forest land and non-forest land, and the results show that the classification of forest land and non-forest land for different years using an NDVI value of 0.62 was satisfactory. Moreover, a total of 2352 training samples were used for model training in this study. Different years used the training samples of the corresponding (different) years.

Accuracy validation

What about the accuracy for other yearsï¼ Yes, I agree that the collection of validation samples is difficult in earlier years? But you only classify forest and non-forest, the collection of samples directly from Landsat images is also possible in the early years.

**Reply:** We selected 987 randomly distributed sampling points from the GF-2 images acquired in 2015 for the accuracy validation. The overall accuracy was found to be 90.37%, and the F1-scores for the broad-leaved, coniferous forest and non-forest land were 0.85, 0.93 and 0.91, respectively. Considering the consistency of the Landsat series of images, the above validation was still considered to be valid for the earlier years because it is difficult to obtain the measured data or the high-resolution satellite images of the study area for these times.

It is an interesting attempt to collect samples directly from early Landsat images for accuracy validation. We collected 584 sample points from 1985 Landsat images for accuracy validation, and the overall accuracy in 1985 was 89.04%, with F1 scores of 0.87, 0.89 and 0.91 for broadleaf, coniferous and non-forested forests, respectively. Considering the potential uncertainty of the sample collection process, the classification accuracy in 1985 is acceptable.

Results and discussion

Overall, the results and discussion are overloaded with descriptions of the methods and results, but lack analysis of the results. It is not recommended to put the results and discussion together. The discussion needs to be improved. The results need to be better integrated with the context of the study. E.g., what is the significance of your results? Rather than simply describing the results.

**Reply:** Thanks. According to your suggestions, we have improved the discussion to better integrate it with the context of the study and to highlight the research significance of this work. The information has been added in this manuscript.

All above revisions are highlighted in the manuscript information. We hope you will be satisfied with our changes. Thanks again for your good suggestions.

Specifically,

1. Where can I find the classification accuracy?

**Reply:** We selected 987 randomly distributed sampling points from the GF-2 images acquired in 2015 for the accuracy validation. The overall accuracy was found to be 90.37%, and the F1-scores for the broad-leaved, coniferous forest and non-forest land were 0.85, 0.93 and 0.91, respectively. (Line 150-152)

2. Did the authors ever consider the collinearity between climate variables? From Fig 10, the curve of TEM_Growmax is very similar to the curve of TEM_Grow.

**Reply:** Thanks for your careful suggestion. As you mentioned, the climate variables selected in this study are to some extent interrelated, which may lead to over-fitting of the regression model. Therefore, we chose a partial least squares regression model for assessing the response of forest change to climate variables. The PLS regression method is a robust multivariate technique that combines features of principal component analysis and multiple regression (Abdi, 2010) and is more parsimonious and statistically robust than principal components regression (Smoliak et al., 2015). Moreover, PLS regression can effectively deal with the problem of multicollinearity (Hou et al., 2020). So, PLS regression is particularly suitable for our case. The information has been added in this manuscript. (Line 310-314)

3. From Fig 11, it seems like the relationship between the forest cover and species changes and climate variables was conducted at a regional scale, not a pixel scale. Why?

**Reply:** The objectives of this work were to quantify the spatiotemporal changes occurring in the boreal forests in Siberia and then to find which climate factor was the main driver of these changes. At the regional scale, the partial least squares regression models successfully presented the response of boreal forest cover and species change to climate variables, and confirm that temperature is the main climatic factor driving the change. The holistic response of boreal forests to climate variables at the regional scale will enable a macroscopic understanding of trends in boreal forests under the context of climate change. Smaller scales, e.g., a pixel scale, the response of boreal forests to climate change needs more of our efforts to be studied.

4. Why did not contain coniferous forests in Fig 11?

**Reply:** Thanks. We have also conducted relevant studies on coniferous forests. However, the results show that coniferous forests are less sensitive and regular in response to climate change compared with broad-leaved forests. And the purpose of this study is mainly to focus on the changes of broad-leaved forests in Siberia. Therefore, coniferous forests are not included in Figure 11.

5. Where is the $Q^2$ value for each PLS regression model?

**Reply:** The $Q^2$ values for the effect of the climate variables on the forest cover and species changes in the PLS regression models for the 5–year, 10–year, and 15–year time intervals were 0.22, 0.21, and 0.20, respectively. (Line 331-333)

6. The correlation coefficients in Table 5 are too low to account for the impact of climate on forest cover changes. In the text Line, 289 is correlation coefficients, and the title of Table 5 is

standardized regression coefficients, please check regression coefficients and correlation coefficients are not the same thing and should not be used in confusion, this puzzled me.

**Reply:** We have revised the "regression coefficients" to "standardized regression coefficients" in the paper. (Line 330)

7. It is possible to compare whether the local temperature exceeds the vegetation's optimum growth temperature.

**Reply:** Thanks for your valuable suggestion. Exploring the dynamics of optimal growth temperature of vegetation under climate change is one of our current ongoing works.

Thanks again for your constructive comments.

**Reference**

Abdi, H.: Partial least squares regression and projection on latent structure regression (PLS Regression). Wiley Interdiscip. Rev. Comput. Stat., 2(1), 97–106, https://doi.org/10.1002/wics.51, 2010.

Aitken, S.N., Yeaman, S., Holliday, J.A., Wang, T., and Curtis-McLane, S.: Adaptation, migration or extirpation: climate change outcomes for tree populations. Evol. Appl., 1(1), 95–111, https://doi.org/10.1111/j.1752-4571.2007.00013.x, 2008.

Balshi, M.S., Mcguire, A.D., Duffy, P., Flannigan, M., Kicklighter, D.W., and Melillo, J.: Vulnerability of carbon storage in North American boreal forests to wildfires during the 21st century. Glob. Change Biol., 15, 1491–510, https://doi.org/10.1111/j.1365-2486.2009.01877.x, 2009.

Barichivich, J., Briffa, K.R., Myneni, R.B., Osborn, T.J., Melvin, T.M., Ciais, P., Piao, S. and Tucker, C.: Large-scale variations in the vegetation growing season and annual cycle of atmospheric CO2 at high northern latitudes from 1950 to 2011. Glob Change Biol., 19, 3167–3183, https://doi.org/10.1111/gcb.12283, 2013.

de Groot, W.J., Flannigan, M.D., and Cantin, A.S.: Climate change impacts on future boreal fire regimes. Forest Ecol. Manage., 294, 35–44, https://doi.org/10.1016/j.foreco.2012.09.027, 2013.

Kasischke, E.S., and Turetsky, M.R.: Recent changes in the fire regime across the North American boreal region—spatial and temporal patterns of burning across Canada and Alaska. Geophys. Res. Lett., 33, L09703, https://doi.org/10.1029/2006GL025677, 2006.

Loarie, S.R., Duffy, P.B., Hamilton, H., Asner, G.P., Field, C.B., and Ackerly, D.D.: The velocity of climate change. Nature, 462(7276), 1052–1055, https://doi.org/10.1038/nature08649, 2009.

McKenney, D.W., Pedlar, J.H., Lawrence, K., Campbell, K., and Hutchinson, M.F.: Potential impacts of climate change on the distribution of North American trees. BioScience, 57(11), 939–948, https://doi.org/10.1641/B571106, 2007.

McLachlan, J.S., Clark, J.S., and Manos, P.S.: Molecular indicators of tree migration capacity under rapid climate change. Ecology, 86, 2088–2098, https://doi.org/10.1890/04-1036, 2005.

Mitchell, T. D. & Jones, P. D. An improved method of constructing a database of monthly climate observations and associated high-resolution grids. Int. J. Climatol., 25, 693–712, https://doi.org/10.1002/joc.1181, 2005.

Piao, S., Ciais, P., Friedlingstein, P., Peylin P., Reichstein, M., Luyssaert, S., Margolis, H., Fang, J., Barr, A., Chen, A., Grelle, A., Hollinger, D., Laurila, T., Lindroth, A., Richardson, A., and Vesala, T.: Net carbon dioxide losses of northern ecosystems in response to autumn warming. Nature, 451, 49–52, https://doi.org/10.1038/nature06444, 2008.

Price, D.T., Alfaro R.I., Brown, K.J., Flannigan, M.D., Fleming, R.A., Hogg, E.H., Girardin, M.P., LakustaT., JohnstonM., McKenneyD.W., PedlarJ.H., StrattonT., SturrockR.N., ThompsonI.D., Trofymow, J.A., and Venier, L.A.: Anticipating the consequences of climate change for Canada's boreal forest ecosystems. Environ. Rev., 21, 322–365, https://doi.org/10.1139/er-2013-0042, 2013.

Smoliak, B.V., Wallace, J.M., Lin, P., and Fu, Q.: Dynamical adjustment of the northern hemisphere surface air temperature Field: methodology and application to observations. J. Climate, 28, 1613–1629, https://doi.org/10.1175/JCLI-D-14-00111.1, 2015.

Sulla-Menashe, D., Woodcock, C., Friedl, M.: Canadian boreal forest greening and browning trends: an analysis of biogeographic patterns and the relative roles of disturbance versus climate drivers. Environ. Res. Lett, 13, 014007, https://doi.org/10.1088/1748-9326/aa9b8, 2018.

Tian, L. Fu, W.: Bi-Temporal Analysis of Spatial Changes of Boreal Forest Cover and Species in Siberia for the Years 1985 and 2015. Remote Sens., 12, 4116, https://doi.org/10.3390/rs12244116, 2020.

Veraverbeke, S., Rogers, B., Goulden, M., Jandt, R., Miller, C., Wiggins, E., and Randerson, J.: Lightning as a major driver of recent large fire years in North American boreal forests. Nature Clim. Change, 7, 529–534, https://doi.org/10.1038/nclimate3329, 2017.

White, J., Wulder, M., Hermosilla, T., Coops, N., Hobart, G.: A nationwide annual characterization of 25years of forest disturbance and recovery for Canada using Landsat time series, Remote Sens. Environ., 194, 303–321, https://doi.org/10.1016/j.rse.2017.03.035, 2017.

Zhu, K., Woodall, C.W., Ghosh, S., Gelfand, A.E., and Clark, J.S.: Dual impacts of climate change: forest migration and turnover through life history. Global change biolo., 20(1), 251–264, https://doi.org/10.1111/gcb.12382, 2013.

---

## Author Comment (AC2)

**Response to Anonymous Referee #2 Comments**

Dear Editors and Referee:

Thanks for your kind comments concerning our manuscript entitled "Spatiotemporal changes in the boreal forest in Siberia over the period 1985–2015 against the background of climate change". Those comments are all valuable and very helpful for improving our paper, and we have made correction which we hope meet with approval. Revised portion are marked in the manuscript. The main corrections in the paper and the responds to the reviewer's comments are as flowing:

Anonymous Referee #2:

The manuscript by Fu et al., entitled "Spatiotemporal changes in the boreal forest in Siberia over the period 1985–2015 against the background of climate change" presents a change analysis in forest cover over a broad area in Central Siberia. The changes in forest are then examined in the context of climate change. The analysis is based on time series of Landsat TM images that are validated with high resolution images from a satellite instrument named Gaofen-2. The specific research questions ask 1) what the extent of change in forest cover and proportions of tree species are, 2) at what latitude forest cover and tree species are most sensitive to climate change, and 3) which climate change factor is the main driver influencing the observed changes. The main findings state that the total forest cover increased over the study period, with coniferous and broadleaved forests showing different patterns of change at different latitudes. The authors identified the forest changes are driven mostly by temperature instead of precipitation.

As the authors state, global change is especially affecting and driving changes in the high latitude boreal forests. These changes are less often studied in boreal forests of northern Eurasia compared to, e.g., North America. Hence, the premise of the study is interesting and the topic merits examination. The study covers a broad area, and the quality of the technical work is very high. However, I do not think that all the conclusions of the study are credible. This concern is especially related to research question three that examines the connection between the climatic variables and forest changes. In my opinion, the changes observed in forest cover and the proportion of deciduous and coniferous forest are too easily attributed to climate change. Furthermore, the role of natural forest dynamics and disturbances is completely disregarded in the manuscript although they could have a strong explanatory power on the observed changes. I think the results need to be interpreted also in this context! I also think that the quality of the study could be increased with a more thorough examination of the used remote sensing data. For example, what is the proportion of non-forest land turning into forest dominated by broadleaved forest during the study period, what influence do major disturbances such as fire have on the results? I think the material used in this study (i.e., Landsat images) would also enable considering the role of forest disturbances. As a second major concern, I think that the text needs a thorough revision to enhance its readability. These and my other concerns are described in more detail below.

**Major comments:**

1. I am not convinced that the observed increase in the proportion of broadleaved forest is solely driven by changes in temperature – correlation does not necessarily mean causation. The natural

succession of Siberian boreal forests typically follows a pathway where broadleaved trees or *Larix spp.* dominate the early successional stage. In later successional stages the broadleaved trees with short longevity are often replaced by conifers such as *Pinus spp.* and *Picea spp.* if stand-replacing disturbances remain absent. The disturbance regime of the area examined in this study includes both ground and crown fires, the latter occurring especially in young coniferous forests that also have woody debris that acts as a fuel load. According to Kharuk et al. 2021 (https://doi.org/10.1007/s13280-020-01490-x) and fire maps complied by Global Forest Watch, large fires have occurred in the studied area, even during the study period. I would be interested to know what is the role of these disturbances and forest dynamics in, e.g., explaining the observed increases in the proportion of broadleaved trees? Similarly, cessation of anthropogenic land use, such as abandonment of agricultural land and clearcutting without reforesting may start a natural succession and increase the proportion of broadleaved trees (as hinted by the authors at L. 325). I think the role of these processes should be examined prior stating that the observed changes are driven by climate change. Maybe you could use the Landsat images to quantify the rate at which non-forest land converted to broadleaved or conifer forest during the study period, and check whether disturbances could explain some of this conversion (see https://doi.org/10.1016/j.rse.2019.111403)? Forest fire and climate are also interlinked as rising temperatures and potentially declining precipitation may increase the fire prevalence in Siberia. This could also be considered in the paper.

**Reply:** Thank you for your constructive suggestions. We agree with you that wildfire or natural succession could have an impact on changes in forest cover and forest types. This paper mainly considers the role of human factors and natural factors in the temporal and spatial changes of forests, and we consciously chose regions with less human activities, especially in high latitudes. Forest fire is indeed an important influencing factor of forest change, and we treat it as a natural factor in the research area with less human activities. In a large time space range, forest fire is also closely related to climate change, just as the referee mentioned, so the unsteady impact due to forest fire over a long period of time is also considered as part of the background of climate change in this paper.

We have added a discussion of the effects of wildfire and natural succession on changes in forest cover and forest types. Moreover, the influence of climatic variables on forest fires was also considered.

Line 268-281
There are a variety of evidence points to complex connections (and changes) in the relationship between disturbance regimes and climate change in boreal forest (Kasischke and Turetsky 2006; Balshi et al., 2009; de Groot et al., 2013). In particular, studies have found that warming and drying trends in Canada's boreal regions favor higher frequency of both fire and insect disturbance (Sulla-Menashe, et al., 2018). While in Siberia, Warming has led to an increase in the frequency and area of wildfires that have reached the Arctic Ocean shore, which is the most important factor in taiga dynamics; furthermore, larch and Scots pine have evolved under conditions of periodic forest fires, thereby gaining a competitive advantage over non-fire adapted species (Kharuk et al., 2021), which may affect forest cover and forest type change in the region. It can be seen from Figure 9(c) that, in the zone 57 ˚N–63 ˚N, $\Delta R_{bf}$ is above 3.8%. Meanwhile, as discussed above, the

absolute increase given by $\Delta R_{cl}$ is negative whereas $\Delta R_{bl}$ is positive in this zone. Also, as $R_{bf}$ had a value of 35.08% in 2015 in the zone 57°N–59°N, at the current rate of change, broad-leaved forest will replace coniferous forest as the dominant tree species in this zone in about 120 years. In general, species will be more resilient at the centers of their present-day distributions, while changes in succession and species composition will be most rapid at the boundaries. Based on current knowledge, the boreal climate zones are expected to shift 5–10 times faster than the speed of natural range expansion achievable by most tree species (McLachlan et al., 2005; McKenney et al., 2007; Aitken et al., 2008; Loarie et al., 2009).

Line 286-293

Previous studies have shown that early northward colonization of tundra ecozones may be dominated by black and white spruces, which are often already established at the treeline. Where soil conditions permit (or where they are improving as a result of warming and drying), air-borne seeds from birch and aspen are likely to arrive and germinate success fully, leading gradually to a forest with significantly greater deciduous content (Price et al., 2013). However, in the 51°N–61°N and 67°N–69°N zones, the values of $\Delta R_{bl}$, $\Delta R_{cl}$ and $\Delta R_{bf}$ are relatively stable, which shows that the rate of increase in these forest type did not change much over the period studied. Therefore, the key to the validity of the response of boreal forests to climate change is to determine whether climate warming is driving significant expansion beyond the present-day forest extent, or faster stand growth and replacement (Zhu et al., 2013).

Line 384-389

Additionally, forest fires and climate are interrelated, and increasing temperatures and potential decreases in precipitation possibly increase the frequency of wildfires in Siberia, which will inevitably result in changes in forest cover and forest type dynamics (kuaruk et al., 2021). Most importantly, given the projected rate of climate change in the Siberian boreal forest, continued research is necessary to more fully understand how future changes in temperature and precipitation regimes in the boreal region will affect coupled patterns of forest cover and forest type change in this vulnerable, geographically extensive biome.

2. How were coniferous and broadleaved trees separated in the study? According to the paragraph title this should be explained in the paragraph starting at L. 110, but I do not find the information from therein. According to Hovi et al. 2017 (https://doi.org/10.14214/sf.7753), *Larix spp.* and certain deciduous trees have similar spectral characteristics. Is it possible that certain *Larix spp.* stands were falsely classified as broadleaved trees, influencing the obtained results?

**Reply:** Thanks for your careful suggestion. We used a hierarchical classification method for forest cover and forest species, i.e., we first used decision trees to distinguish between forested and non-forested land, and then used a random forest approach to classify coniferous and broadleaved forests. This can effectively avoid the influence on the final classification results due to same object with different spectrums and same spectrum of different objects (e.g., *Larix spp.* and certain deciduous trees have similar spectral characteristics (Hovi et al., 2017)).

Line 158-162

We selected 987 randomly distributed sampling points from the GF-2 images acquired in 2015 for the accuracy validation. The overall accuracy was found to be 90.37%, and the F1-scores for the

broad-leaved, coniferous forest and non-forest land were 0.85, 0.93 and 0.91, respectively. Considering the consistency of the Landsat series of images, the above validation was still considered to be valid for the earlier years because it is difficult to obtain the measured data or the high-resolution satellite images of the study area for these times.

It is an interesting attempt to collect samples directly from early Landsat images for accuracy validation. We collected 584 sample points from 1985 Landsat images for accuracy validation, and the overall accuracy in 1985 was 89.04%, with F1 scores of 0.87, 0.89 and 0.91 for broadleaf, coniferous and non-forested forests, respectively. Considering the potential uncertainty of the sample collection process, the classification accuracy in 1985 is acceptable.

3. I would be interested to know if the observed changes in forest cover are due to forests with open canopy structure becoming denser or conversion of previously non-forested land into forests? These two processes have very different significance for forest dynamics. Including results of this examination in the study would increase its information content and general interest.

**Reply:** Thanks for your suggestion. The change in forest cover over the study time period was due to the conversion of previously non-forested land to forested land. We carefully checked the classification results for forested and non-forested lands to ensure the reliability of the results (overall accuracy of 90.37%).

**Minor comments:**

L. 1 (title): In the manuscript text the authors state that the study area ranges from temperate to boreal forests. If that is the case, I suggest rephrasing the title of the study.

**Reply:** Thanks. According to Brandt (2009), the boreal zone was defined as the broad, circumpolar vegetation zone of high northern latitudes covered principally with forests and other wooded land, includes the temperate zone. Therefore, we contained the temperate zone in the boreal forest in this paper.

L. 10: At a faster rate compared to where?

**Reply:** Thanks. Climate change has been proven to be an indisputable fact and to be occurring at a faster rate (compared to the other regions of the world) in boreal forest areas. We have revised the relevant expressions. (Line 10-11)

L. 11 – 12: I would argue that there is quite a lot of evidence on how the climate change is changing boreal forests. This view is shared by the authors as the introduction at L. 45 states that "There has been much research on the effect of climate change on boreal forest". I think that in this context there has been limited focus on Central Russian boreal forests. Please rephrase the introduction.

**Reply:** Thanks for your valuable suggestion. According to your suggestion, we have added research on how the climate change is changing the boreal forests in central Russia.

Line 52-66

Over the past 30 years, spring and autumn temperatures over northern latitudes have increased by about 1.1 °C and 0.8 °C, respectively (Mitchell and Jones 2005), and the thermal potential growing season has lengthened by about 10.5 days (Barichivich et al., 2013). Several studies indicate that increasing warming may result in accelerating the northward expansion of boreal

forests (Veraverbeke et al., 2017), as well as the observation of a greening trend characterized by a longer growing season and greater photosynthetic activity (Piao et al., 2008). Shuman et al. (2011) showed that climate warming may convert Siberia's deciduous larch (*Larix spp.*) to evergreen conifer forests, and thus decrease regional surface albedo; At the continental scale, when temperature is increased, larch-dominated sites become vulnerable to early replacement by evergreen conifers. Ratcliffe et al. (2017) investigated a forested peatland in western Siberia and showed that climate change has caused the expansion of forested peatlands and increased tree cover. In addition, it is highly probable that the annual mean temperature in Canada's boreal forest region will increase by at least 2 ℃ by 2050 in this century, which may lead to effects on the ecological functioning of the region's boreal forests, such as triggering a process of forest decline and re-establishment lasting several decades, while also releasing significant quantities of greenhouse gases that will amplify the future global warming trend (Price, et al., 2013).

L. 15: At first, I did not know what was meant by the term "forest species", but after reading on I understood that they refer to broadleaved and coniferous forests. I would not call these "forest species" but forest types. I would change "forest species" to "forest type" and explicitly say that by forest type you mean coniferous and broadleaved forests.
**Reply:** Based on your suggestion, we have revised the term "forest species" to "forest type".

L. 31 – 32: What is "geographical footprint"? Largest area?
**Reply:** Thanks. The sentence means that the boreal forest biome has one of the largest geographic footprints of any terrestrial biome on the planet (Olson et al. 2001). (Line 31-32)

L. 32: Remove "and encircles the globe at northern latitudes" as circumpolar distribution is already mentioned at L. 30.
**Reply:** Thanks. We have deleted the sentence.

L. 32 – 35: I find the ending of this sentence paradoxical. It reads that research of range shifts in boreal forests has focused on species-specific responses on temperate tree species (i.e., different biome). Please rephrase what is meant by this.
**Reply:** Thanks. We have rewritten the sentence that "To date, research into shifts in the range of this biome has predominately focused on the advance of boreal tree species into tundra or alpine habitats (i.e., treeline advance; see Harsch et al. 2009), or on the species-specific responses of temperate tree species (Zhu et al. 2012)". (Line 32-34)

L. 40 – 41: Could you be a bit more specific – what kind of changes in biodiversity are expected due to climate change?
**Reply:** Thanks. We have added the corresponding expression.
Changes to biodiversity are one of the expected responses to climate change, for example, some of the most important conifer species in British Columbia are expected to lose a large portion of their suitable habitat (Hamann and Wang, 2006). (Line 39-41)

L. 42 – 44: The message of this sentence is very difficult to understand. Please clarify the point that the sentence tries to make.

**Reply:** Thanks. We have rewritten the sentence based on your suggestion.

Most importantly, climate change is expected to reduce climatic constraints on plant growth (Nemani et al, 2003): warmer, wetter conditions will result in increased vegetation productivity, which has been shown to be an indirect indicator of biodiversity, correlated with geographic variation in species richness (Coops et al., 2008; Nelson et al., 2014). (Line 42-45)

L. 46 – 47: Could you give an example of the spatiotemporal differences in tree growth in the boreal biome that are caused by climate change?

**Reply:** Thanks. We have added the corresponding expression.

However, there are clear spatiotemporal differences in these effects (Alibakhshi et al., 2020). For example, Hou et al. (2020) found that vegetation phenology indicators in Finland's boreal forests showed spatiotemporal differences in response to climate variables in different months, i.e., vegetation in different regions showed different patterns of response to climate variables. (Line 48-50)

L. 49: White spruce (Picea glauca) is a species that is native to boreal North America. Changes in growth of white spruce are then hardly relevant in the context of Eurasian boreal forest, right? Could this be replaced with an example from the same study region?

**Reply:** Thanks for your careful suggestion. This paragraph is to support the statement that "It has been observed that the growth of boreal forest has been influenced by global warming in the past decade or more". Based on your suggestion, we have added the example of larch (*Larix spp.*) in Siberia.

Line 57-59

Shuman et al. (2011) showed that climate warming may convert Siberia's deciduous larch (*Larix spp.*) to evergreen conifer forests, and thus decrease regional surface albedo; At the continental scale, when temperature is increased, larch-dominated sites become vulnerable to early replacement by evergreen conifers.

L. 59 – 61: This statement is not true. See, e.g., https://doi.org/10.1007/s10980-020-00979-w on the use of multispectral aerial photographs for this purpose.

**Reply:** Thanks for your careful suggestion. We have revised the statement that the Landsat series data are the most widely used multispectral dataset for monitoring natural and human-induced landscape changes at the scale of tens of meters over periods of years or decades (Matasci et al., 2018; Hadi et al., 2016; Hermosilla et al., 2019). (Line 73-76)

L. 70 – 71: What is meant by "from the temperature to the frigid zones"?

**Reply:** Thanks. This word "temperature" is a misnomer, we revised "temperature" to "temperate". In addition, Brandt (2009) defined the boreal zone as the broad, circumpolar vegetation zone of high northern latitudes covered principally with forests and other wooded land, includes the temperate zone. The study area of this work, Krasnoyarskiy Kray, encompasses the temperate to frigid zones.

L. 83 (Fig. 1): An inset map would be helpful in locating the study region.

**Reply:** According to your suggestion, we revised Figure 1 to locate the study area.

[Figure]

[Figure]

**Figure 1.** Location of the study area together with the DEM and false-color composite of Landsat 8 images.

L. 111 – 118: Why is discriminating forested/vegetated areas from non-vegetated areas presented twice in these paragraphs?

**Reply:** Thanks for your careful suggestion. We have revised the paragraph to correctly express that "Finally, a random forest (RF) algorithm was used to discriminate coniferous and broadleaved forests from areas of vegetation (Breiman 2001; Strobl et al., 2007; Cutler et al., 2008; Svetnik et al., 2003; Rodriguez-Galiano et al., 2013; Assiri 2021; Climent et al., 2019)". (Line 150-152)

L. 126: Did you visually classify the sampling points based on GF-2 images?

**Reply:** Yes. In this study, we selected the sample points used for the classification based on Landsat images refer to GF-2 images and Google Earth images (Gong et al., 2013).

L. 153 – 154: That the forest cover has not changed much over the study period does not mean that the forests are not significantly affected by climate change but that the influence does not

manifest as changes in forest cover. Please rephrase.

**Reply:** Thanks. We have rewritten the sentence that "Taking into account the accuracy of the forest cover retrieval, it can be considered that the forest coverage in this zone has not changed over the study period, which means that the cover of boreal forest in this zone has not been significantly affected by climate change".

L. 157 – 158: Similar to the previous comment, from the fact that the strongest change in forest cover was observed in the northernmost zone it does not follow that the forests in these areas were mostly affected by climate change (see also my comment on forest dynamics). Rephrase "The fastest change was observed in the northern zone from 63 °N–69 °N, which means that this is the area where the forest has been most affected by climate" for example to "The fastest change was observed in the northernmost zone (63 °N–69 °N) that is also the zone where the climate warming is also projected to be the highest".

**Reply:** Thanks. According to your suggestion, we have rewritten the sentence that "The fastest change was observed in the northernmost zone (63 °N–69 °N) that is also the zone where the climate warming is also projected to be the highest".

L. 187: Change to "not found north of the latitude 67°N in the studied region.

**Reply:** Thanks. we have rewritten the sentence that "In the northernmost zone (67 °N–69 °N), $R_{cl}$ is still above 25% whereas $R_{bl}$ is only about 5%, which indicates that coniferous forest is more resistant to cold and that broad-leaved forest is essentially not found north of latitude 67 °N in the studied region".

L. 202: Change "increase" to "change" as the forest cover may also decrease, right?

**Reply:** Yes. Based on your suggestion, we have revised the word "increase" to "change".

L. 217 – 219: As I have stated in my previous comments, the fact that the coverage of broadleaved forests has increased in the study region is not necessarily only due to climate change. Please revise.

**Reply:** Thanks. We have rewritten the sentence that "Overall, it can be seen that the broad-leaved forest coverage increased in every latitude zone, which means that the climate change that has been occurring may have promoted the growth of broad-leaved species across the study area during the three decades of the study".

L. 228 – 229: Also, other drivers than climate change may explain the decrease in the cover of coniferous forest. Please revise.

**Reply:** Thanks. We have rewritten the sentence that "However, the area of coniferous forest in the medium latitude zone 57 °N–63 °N has declined slightly over the three decades of the study with a value of $\Delta R_{cl}$ of about –2.3%; in comparison $\Delta R_{bl}$ is about 3.9%, which means that climate change may have had a negative impact on coniferous forest growth in this zone".

L. 254 – 255: See the previous two comments.

**Reply:** Thanks. We have rewritten the sentence that "Given that the amount of human activity in the study area is limited, it is reasonable to assume that the changes in the forest may be driven

mainly by climate variables".

L. 318 – 320: For me it is uncertain if the author suggest that the results indicate a northwards shift in the range of temperate biome. Such range shifts occur during longer time scales than those considered in this study. Please clarify what is meant by this sentence.

**Reply:** Thanks. This sentence is to support the statement that "In addition, it can be seen from Figure 12(d) that $\Delta R_{bl}$ responds positively to an increase in temperature, which indicates that broad-leaved forest is sensitive to warming and benefits from higher temperatures", not to suggest that "the northwards shift in the range of temperate biome".

L. 319: Replace "temperature" with "temperate".

**Reply:** Based on your suggestion, we have revised the word "temperature" to "temperate".

L. 320: Are you suggesting that the southernmost forests of the study region are transforming into subtropical or tropical forests? Please rephrase.

**Reply:** Thanks. We have rewritten the sentence that "Moreover, it has been found that the distribution of many forest types is beginning to expand towards the poles, with temperate forests gradually shifting into areas previously covered by boreal forests; the southern parts of these temperate forests will be replaced by subtropical or tropical forests. The northern boundary of temperate forests is also shifting towards the poles (Hirota et al., 2010; Parmesan and Yohe, 2003)".

L. 324: Also other drivers than climate change may explain the identified patterns.

**Reply:** Thanks. We have rewritten the sentence that "Furthermore, warming has a positive effect on $\Delta R_{bl}$, meaning that the observed increase in $\Delta R_{bl}$ was may be caused by a rise in temperatures".

L. 333 – 334: The term "significant difference" refers to differences that are statistically significant. However, the significance of the changes in forest cover was not statistically tested in the study. Please rephrase.

**Reply:** Thanks for your careful suggestion. we have rewritten the sentence that "In this study, changes in the area of the Siberian boreal forest and the forest species in Krasnoyarskiy Kray, Russia, were quantified using remote sensing data covering the period 1985 to 2015. The results show that there are differences in the changes that were observed across the study area".

L. 335 – 336: That the forest cover changed increased the most in the northernmost latitudes does not automatically mean that this region is the most sensitive to climate change. Instead, these forest might be – for example – recovering from a major disturbance. Please revise.

**Reply:** We have rewritten the sentence that "Overall, the total forest area increased continuously over the three decades of the study, particularly in the high-latitude part of the study area, which may indicate that the boreal forest in this region is the most sensitive to climate change".

All above revisions are highlighted in the manuscript information. We hope you will be satisfied with our changes. Thanks again for your good suggestions.

**References**

[revised manuscript text omitted]

---

## Author Response (AR2)

**Response to Anonymous Referee #1 Comments**

Dear Editors and Referee:

Thanks again for your kind comments concerning our manuscript. Those comments are all valuable and very helpful for improving our paper, and we have made correction which we hope meet with approval. Revised portion are marked in the manuscript. The main corrections in the paper and the responds to the reviewer's comments are as flowing:

**Anonymous Referee #1:**

Q 1: The article has improved a lot. But I still have concerns about the method for the classification of forested land areas. The authors used a threshold-based method (or decision tree) to extract the forested areas. They used constant thresholds of NDVI (0.61), RVI (6), and NIR (0.38) to extract the forest areas for the different years. This is questionable. Firstly, the Landsat TM and Landsat OLI data are inherently slightly different (wavelengths in each band); secondly, the acquisition time of the images is not consistent from year to year, some in the summer and some in October; and finally the Qin, et. al. (2015) may be a wrong reference and I did not find that they used the same method. I would expect slightly varying thresholds over years. A better approach is to use the RF model directly to classify coniferous forests, broadleaf forests, and non-forestry forests for each year.

**Reply:** Thanks for your careful suggestion. Qin et al. (2015) showed that the annual maximum NDVI ($NDVI_{max}$) values of built-up areas, barren lands, and sparsely vegetated lands are usually lower than 0.30, whereas forest $NDVI_{max}$ values are usually higher than 0.50. However, empirical NDVI values for different times and regions are not universal (Ma et al., 2019). For this study area, we determined the decision tree classification rules based on sample training: NDVI values greater than 0.62 and RVI values greater than 6.0 were selected as vegetation land, otherwise land was regarded as non-vegetation land. According to your suggestion, we have added the methodological limitations in the discussion section.

Line 142-151

In this classification, two vegetation indices, NDVI and ratio vegetation index (RVI), were used for the discrimination between forest and nonforest land. NDVI can effectively weaken the effects of complex terrain in image information extraction, and enhance the distinction between vegetation and other land types, which is helpful for improving the accuracy and credibility of forest information extraction. The RVI can better reflect the difference of vegetation growth and coverage, and is suitable for vegetation monitoring in areas with vigorous vegetation growth and high coverage. The annual maximum NDVI ($NDVI_{max}$) values of built-up areas, barren lands, and sparsely vegetated lands are usually lower than 0.30, whereas forest $NDVI_{max}$ values are usually higher than 0.50 (Qin et al., 2015). Subsequently, we determined the decision tree

classification rules based on sample training: NDVI values greater than 0.62 and RVI values greater than 6.0 were selected as vegetation land, otherwise land was regarded as non-vegetation land.

Line 185-188
However, empirical NDVI values for different times and regions are not universal (Ma et al., 2019). Therefore, the results of the threshold-based method (decision tree) used in this study for the classification of forested and non-forested land may have a certain uncertainty. Future research could use the machine learning or deep learning methods to classify forests to improve the accuracy of classification.

Q 2: For the training sample, I would not agree that it is very difficult to obtain for each year. First of all the forest area has stability, so the samples acquired in 2015, after checking, can be used for other years as well (some samples may need to be added and removed appropriately). This process can be done with the help of Google Earth history high-resolution images, or Landsat images. It is not a challenge for classifying only three categories (Conifer, broadleaf, and non-forest).
**Reply:** Thanks. Yes. As you said training samples can be obtained with the help of Google Earth history high-resolution images, or Landsat images. In this study, we selected the sample points used for the classification for different years based on Landsat images refer to GF-2 images and Google Earth images (Gong et al., 2013). (Line 158-160)

Q 3: Furthermore, the description in the methods section needs further refinement. For example, "it was found that different plants have different spectral reflectance peaks in the near-infrared band" -- I would say the peak reflectance of different plants may be also different in some other spectral regions. This sentence has no meaning.
"This band is highly sensitive to the differences in reflectance that result from different types of leaves having different internal structures" --- not only the structure but also the color, etc. Also missing a reference here.
**Reply:** Thanks for your careful suggestion. According to your comments, we revised the corresponding sentences and added references.

Line 151-154
In this work, it was found that different plants have different spectral reflectance peaks in the near-infrared band; this band is highly sensitive to the differences in reflectance that results from different types of leaves having different internal structures and colors (Lewis 2002).

Q 4: "Six bands, Landsat TM bands 1–5 and 7, and Landsat OLI bands 2–7 were selected as characteristic spectral variables, and meanwhile NDVI, the normalized difference index (NDI) (Rodríguez-Moreno and Bullock, 2014) " --- wrong reference.
**Reply:** Thanks for pointing out this. We revised this error and added a reference.

Line 160-162

Six bands, Landsat TM bands 1–5 and 7, and Landsat OLI bands 2–7, were selected as characteristic spectral variables, and meanwhile NDVI, the normalized difference built-up index (NDBI) (Cha et al., 2003) and the RVI were also selected as index characteristic variables for classification in RF.

Q 5: "The overall accuracy was found to be 90.37%, and the F1-scores for the broad-leaved, coniferous forest and non-forest land were 0.85, 0.93, and 0.91, respectively (Pontius and Millones, 2011)." --- why the result has a reference???

**Reply:** Thanks for your careful suggestion. Previous anonymous referee pointed out that the Kappa may is not a reliable accuracy metric. He recommended to us a very good paper, i.e., Death to Kappa: birth of quantity disagreement and allocation disagreement for accuracy assessment (Pontius and Millones, 2011), and suggested using F1 scores (the harmonic means of the user's and the producer's accuracies) for assessing classification accuracy. This reference is the F1 score citation. we revised the corresponding sentences and added references.

Line 165-166

The overall accuracy was found to be 90.37%, and the F1-scores (Chen et al., 2021; Pontius and Millones, 2011) for the broad-leaved, coniferous forest and non-forest land were 0.85, 0.93, and 0.91, respectively.

Q 6: "Following this, a haze optimized transformation (HOT) algorithm was used to identify and remove noise due to thin clouds" ---- Have you filled the gaps caused by cloud removal? If the cloud pixels that you removed happen to be forested areas, will that affect the results of your change detection?

**Reply:** Thanks for your good suggestion. The haze optimized transformation (HOT) algorithm was used to identify and remove noise due to thin clouds. We used the adjacent date images to fill the gaps caused by thick cloud cover. This has a minor effect on the results of change detection in this study.

Q 7: Please revise the method section carefully by referring to remote sensing-related papers!!!

In summary, I understand that it would be difficult (or might be not necessary) to use new methods. But since the classification products serve as the basis for the subsequent analysis, the methodological limitations must be discussed in the discussion section at least, otherwise, I do not consider it suitable for publication.

**Reply:** Thanks for your constructive suggestion. According to your suggestion, we have added the methodological limitations in the discussion section. (Line 185-188)

All above revisions are highlighted in the manuscript information. We hope you will be satisfied with our changes. Thanks again for your good suggestions.

**References**

Cha, Y., Ni, S.X., and Yang, S.: An Effective Approach to Automatically Extract Urban Land-use from TM Imagery. Natl. Remote Sens. Bull., 1, 37–40, https://doi.org/10.3321/j.issn:1007-4619.2003.01.007, 2003.

Chen, J., Yuan, Z.Y., Peng, J., Chen, L., Huang, H.Z., Zhu, J.W., Liu, Y., and Li, H.F.: DASNet: Dual Attentive Fully Convolutional Siamese Networks for Change Detection in High-Resolution Satellite Images. IEEE J. Stars., 14, 1194–1206, https://doi.org/10.1109/JSTARS.2020.3037893, 2021.

Gong, P., Wang, J., Yu, L., Zhao, Y.C., Zhao, Y.Y., Liang, L., Niu, Z.G., Huang, X.M., Fu, H.H., and Liu, S.: Fine resolution observation and monitoring of global land cover: First mapping results with Landsat TM and ETM+ data. Int. J. Remote Sens., 34, 2607–2654, https://doi.org/10.1080/01431161.2012.748992, 2013.

Lewis, M.: Spectral characterization of Australian arid zone plants. Can. J. Remote Sens., 28(2), 219–230, https://doi.org/10.5589/m02-023, 2002.

Ma, X.P., Bai, H.Y., Deng, C.H., and Wu, T.: Sensitivity of Vegetation on Alpine and Subalpine Timberline in Qinling Mountains to Temperature Change. Forests, 10, 1105, https://doi.org/10.3390/f10121105, 2019.

Pontius, R.G., and Millones, M.: Death to Kappa: Birth of quantity disagreement and allocation disagreement for accuracy assessment. Int. J. Remote Sens., 32, 4407–4429. https://doi.org/10.1080/01431161.2011.552923, 2011.

Qin, Y., Xiao, X., Dong, J., Zhang, G., Shimada, M., and Liu, J.: Forest cover maps of china in 2010 from multiple approaches and data sources: PALSAR, Landsat, MODIS, FRA, and NFI. ISPRS J. Photogramm. Remote Sens., 109, 1–16, https://doi.org/10.1016/j.isprsjprs.2015.08.010, 2015.